# Bringing the uncultivated microbial majority of freshwater ecosystems into culture

Michaela M. Salcher [1] ✉, Paul Layoun[1,2], Clafy Fernandes[1,2], Maria-Cecilia Chiriac[1], Paul-Adrian Bulzu [1], Rohit Ghai [1], Tanja Shabarova [1], Vojtech Lanta[3], Cristiana Callieri[4], Bettina Sonntag [5], Thomas Posch [6], Fabio Lepori[7], Petr Znachor[1,2] & Markus Haber[1]

Axenic cultures are essential for studying microbial ecology, evolution, and genomics. Despite the importance of pure cultures, public culture collections are biased towards fast-growing copiotrophs, while many abundant aquatic prokaryotes remain uncultured due to uncharacterized growth requirements and oligotrophic lifestyles. Here, we applied high-throughput dilution-to-extinction cultivation using defined media that mimic natural conditions to samples from 14 Central European lakes, yielding 627 axenic strains. These cultures include 15 genera among the 30 most abundant freshwater bacteria identified via metagenomics, collectively representing up to 72% of genera detected in the original samples (average 40%) and are widespread in freshwater systems globally. Genome-sequenced strains are closely related to metagenome-assembled genomes (MAGs) from the same samples, many of which remain undescribed. We propose a classification of several novel families, genera, and species, including many slowly growing, genome-streamlined oligotrophs that are notoriously underrepresented in public repositories. Our large-scale initiative to cultivate the "uncultivated microbial majority" has yielded a valuable collection of abundant freshwater microbes, characterized by diverse metabolic pathways and lifestyles. This culture collection includes promising candidates for oligotrophic model organisms, suitable for a wide array of ecological studies aimed at advancing our ecological and functional understanding of dominant, yet previously uncultured, taxa.

Axenic cultures are considered a gold standard in microbiology; however, only a minuscule fraction of the total predicted diversity of $10^6$–$10^{12}$ prokaryotic species[1,2] are so far cultivated[3,4]. The Genome Taxonomy Database (GTDB), a comprehensive database comprising genomes from cultures, metagenome-assembled genomes (MAGs) and single-amplified genomes (SAGs), identified 113,104 species clusters spanning 194 phyla in its latest release (R220)[5]. However, as of May 17, 2024, only 24,745 species from 53 phyla have been validly described under the International Code of Nomenclature of Prokaryotes (ICNP)[3]. Moreover, public culture collections are heavily biased

[1]Institute of Hydrobiology, Biology Centre CAS, Ceske Budejovice, Czech Republic. [2]Faculty of Science, University of South Bohemia, Ceske Budejovice, Czech Republic. [3]Department of Functional Ecology, Institute of Botany of the Czech Academy of Sciences, Průhonice, Czech Republic. [4]Water Research Institute (IRSA) - National Research Council (CNR), Verbania, Italy. [5]Research Department for Limnology, Mondsee, University of Innsbruck, Mondsee, Austria. [6]Limnological Station, University of Zurich, Kilchberg, Switzerland. [7]University of Applied Sciences and Arts of Southern Switzerland (SUPSI), Mendrisio, Switzerland. ✉e-mail: michaelasalcher@gmail.com

towards human-associated microbes[6] or copiotrophs that can easily be isolated and maintained on agar plates and in nutrient-rich media. Copiotrophs are typically rare in nature[7], a phenomenon termed "the great plate count anomaly"[8]. Recent efforts using nutrient-rich agar plates have resulted in copiotrophic cultures capturing up to 45% of the total prokaryotic community of the particle-attached fraction (>20 μm) of the bathypelagic, deep ocean, but only 1.5% of the free-living fraction of the photic ocean[9]. However, the majority of environmental microbes are free-living oligotrophs that are adapted to low nutrient and substrate concentrations common in natural systems such as photic oceans[10], freshwater lakes[11], or subsurface soils[12,13]. In particular, aquatic ecosystems are dominated by very small prokaryotes with reduced genomes (i.e., genome-streamlined)[14] with multiple auxotrophies causing dependencies on co-occurring microbes that supply essential nutrients or detoxify harmful metabolites[15–17]. Further, largely unknown growth requirements, adaptations to low nutrient and substrate concentrations, a tendency of being outcompeted by fast-growing copiotrophs during enrichment, and a free-living lifestyle preventing growth on solid surfaces such as agar plates, make the isolation and cultivation of slowly growing aquatic oligotrophs challenging. Traditional cultivation efforts typically yield strains that contribute only marginally to the natural community in oceans (< 5% in TARA Oceans samples[9,18]) and freshwaters[19]. Dilution-to-extinction cultivation in sterilised sea/lake water media has been suggested as a powerful technique to circumvent the aforementioned problems and has been successfully employed to isolate some of the most abundant aquatic microbes such as members of the SAR11 group[20–22], freshwater *Nanopelagicales*[23,24], *Nitrosopumilaceae*[25], *Methylopumilus* and the marine OM43 clade (*Methylophilaceae*)[26,27] and other diverse prokaryotes[28]. However, autoclaved sea/lake water as cultivation medium has certain drawbacks for stable cultivation, such as seasonally changing nutrient compositions, modifications or cleavage of essential components such as vitamins during sterilisation (e.g., autoclaving) and the access to fresh medium for maintaining cultures[22]. The use of defined, artificial media is therefore preferable to obtain reproducible growth of cultures. Further, it is necessary to characterise the minimal requirements needed for growth, as defined media have to be carefully composed due to different carbon and nutrient requirements of aquatic microbes with diverse and frequently unknown metabolic traits and the risk of growth inhibition by inadequate concentrations (too high or too low) or inappropriate stoichiometry of certain media components[29].

Culture-independent approaches, such as in situ experiments[30,31], genome-centric metagenomics[32,33], or SAGs[34], have revolutionised microbial ecology[35]. The analysis of MAGs and SAGs have uncovered a vast diversity of uncultivated microbes[32,33] and shed light on their occurrence in nature, metabolic traits and evolutionary history[36–38]. However, genomes alone are not sufficient to characterise the ecology of microbial taxa, as many phenotypic traits (e.g., cell size, temperature, pH, salinity, or substrate ranges and optima) are hard to identify or not at all encoded in the genome[39]. Further, cultures are a prerequisite to discover and characterise biochemical pathways[25,40,41], cell ultrastructure[42,43], growth requirements[17,29], and microbial interactions[15,16] and are the basis for genetic manipulations[44,45]. Genomic analyses of so far uncultivated microbes can, however, hint at metabolic requirements and aid in targeted isolation of important taxa via the design of specific cultivation media or through reverse genomics[39,46,47].

Here, we employed a high-throughput dilution-to-extinction approach with three defined artificial media containing either different carbohydrates, organic acids, catalase, vitamins, and other organic compounds in μM concentrations, mimicking carbon concentrations typically found in freshwater lakes (med2 and med3), or methanol, methylamine and vitamins (MM-med) as sole carbon sources (Supplementary Data 1). We sampled 14 lakes in Central Europe during spring and autumn 2019, and four lakes additionally in summer 2019 (Supplementary Fig. 1 and Supplementary Data 2). We took water from the upper water layer (epilimnion, 5 m depth) and from the oxygenated, cold bottom water layer (hypolimnion, variable depth 15–300 m depending on the depth of the respective lake) for cultivation of prokaryotic strains and for metagenomic sequencing of the whole community (*n* = 67; Fig. 1). We aimed to isolate a broad variety of taxa that are abundant in lakes and have so far been underrepresented or missing in culture collections, with a special focus on dominant genome-streamlined bacteria.

## Results and discussion
### High-throughput dilution-to-extinction cultivation of freshwater microbes

A total of 6,144 wells (64 96-deep-well plates) were inoculated with approx. one cell per well for dilution-to-extinction cultivation and incubated for 6–8 weeks at 16 °C. Screening resulted in 1201 initial cultures, of which 229 were identified as mixed by Sanger sequencing of 16S rRNA gene amplicons, and 344 cultures showed no growth after several transfers and were discarded (Supplementary Fig. 2, Supplementary Data 3). A total of 627 cultures were axenic and maintained thereafter (Supplementary Fig. 2). On average, we isolated 10 axenic strains per sample (average viability *V* = 12.6%)[48], with only slightly more cultures gained from the warm epilimnion (57%) than the cold hypolimnion (Fig. 2a). A significantly lower isolation success was observed in spring compared to summer and autumn (*p* = 0.000179 and *p* = 0.00341, respectively, results of statistical tests can be found in Supplementary Data 4), while no difference in viability was found between the three different isolation media. 16S rRNA gene analyses of axenic cultures resulted in 72 distinct genera or alphanumerical SILVA lineages, including some of the most abundant freshwater microbes that yet contain only a few cultured representatives, e.g., *Planktophila*[23,24], *Fontibacterium*[49] (previously proposed as '*Ca.* Fonsibacter')[21], *Methylopumilus*[26,50], undescribed acIV *Acidimicrobiia*[11,28] and a large number of so far uncultivated lineages of *Actinomycetota*, *Pseudomonadota*, *Bacteroidota*, *Verrucomicrobiota*, *Cyanobacteriota*, and *Armatimonadota* (Supplementary Fig. 3)[11]. Our culture collection was mainly dominated by *Pseudomonadota* (50 out of 72 genera, many of them so far uncultivated), however, we failed to obtain isolates from several abundant bacterial (*Chloroflexota*, *Planctomycetota*) and archaeal (*Thermoproteota*) phyla that primarily inhabit the deep hypolimnion of freshwater lakes[11,51]. Only slight differences in the taxonomic composition were recorded for media that contained different types of carbohydrates (med2 and med3, Supplementary Data 1 and Supplementary Fig. 3). The enrichment in a medium with methanol and methylamine as sole carbon sources resulted in an enhanced cultivation of methylotrophs (*Methylopumilus*, *Methylotenera*[50,52]). However, other genera were also able to grow in this C1 medium (e.g., *Polynucleobacter*, *Limnohabitans*) and might have survived by either being photoautotrophic, methylovorous, or by using catalase or vitamins as additional carbon sources.

Most cultures showed stable growth for more than one year, and several (*n* = 18) were characterised in short-term growth assays in up to eight different media (Fig. 2b, Supplementary Data 5 and Supplementary Figs. 4, 5). Although the strains form a continuum in growth characteristics, they could be grouped into oligo- and copiotrophs and a group of strains with in-between features (mesotrophs) after clustering based on growth rates (Supplementary Fig. 6). Oligotrophs affiliated with *Planktophila* (*Actinomycetota*) and the newly proposed genera *Acidimicrobilacustris* gen. nov. (*Actinomycetota*) and *Fimbriicoccus* gen. nov. (*Armatimonadota*) showed slow growth (max. growth rates < 1 d$^{-1}$) and maximum cell yields of < 4 × 10$^7$ cells ml$^{-1}$, similar to previous reports for *Planktophila*[17] and other oligotrophs such as *Fontibacterium*[21,37,49] and *Methylopumilus*[26,37]. No significant

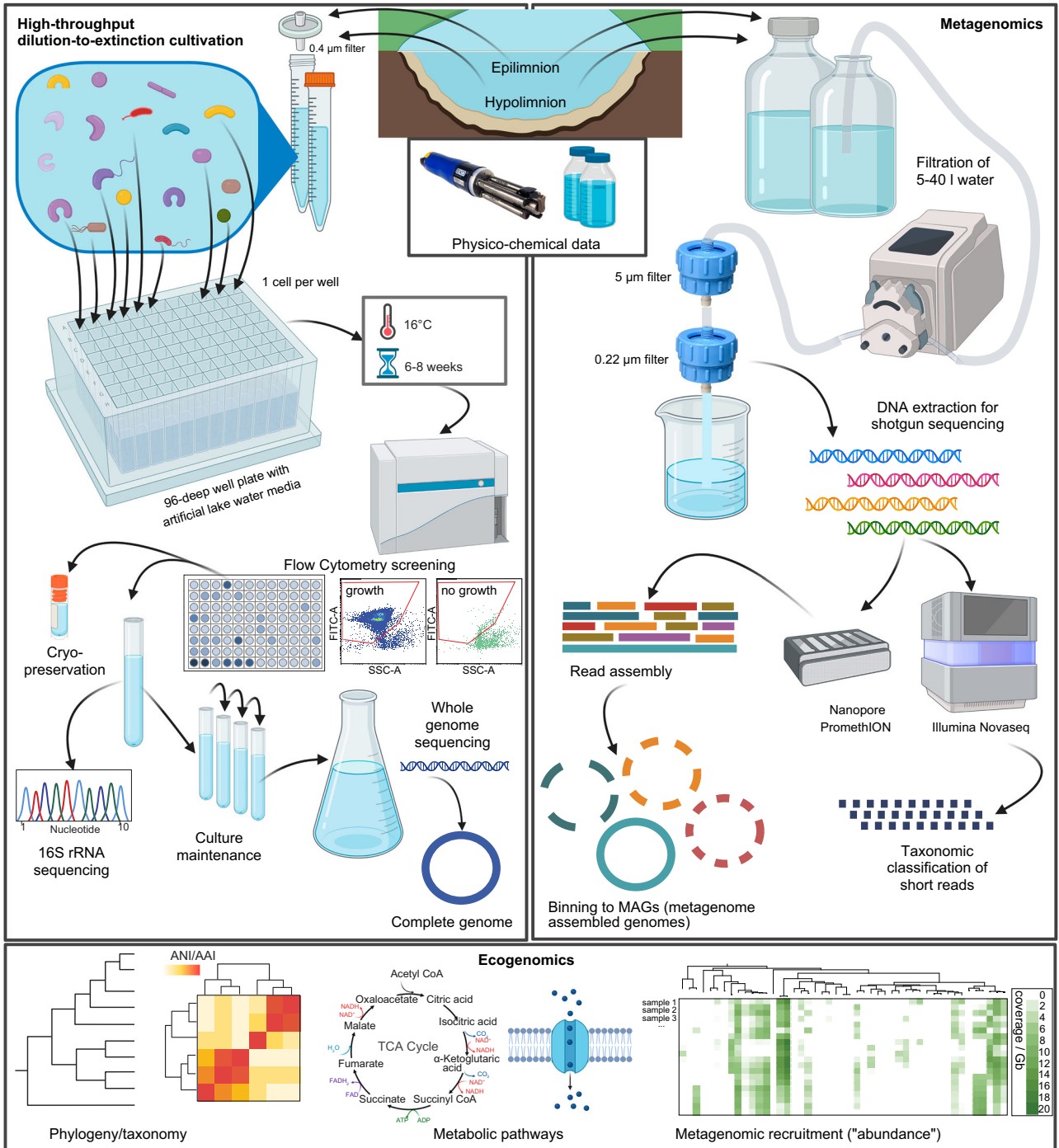

**Fig. 1 | Workflow of sampling, dilution-to extinction cultivation, metagenomic sequencing and ecogenomic analyses.** Created in BioRender. Salcher, M. (2025) https://BioRender.com/gqmisot.

differences in growth rate were observed between media with low carbon content (1.1 and 1.3 mg DOC per litre for med2 and med3, respectively) and high carbon content (10 × med2 and med3, 1:10 and 1:100 diluted NSY [medium containing nutrient broth, soytone, and yeast extract][53] for oligotrophs (Supplementary Data 4).

Highest growth rates were recorded for *Flavobacterium* TH-M1 (6.1 d⁻¹), while many strains displayed growth rates between 2 - 3 d⁻¹ and maximum cell yields of >10⁸ cells ml⁻¹ (*Limnohabitans, Allorhodoferax* gen. nov., *Sphingorhabdus, Polynucleobacter, Rhabdaerophilus*), which have been reported as typical in situ growth rates of copiotrophs in freshwater[54,55] and marine environments[56] as well as in cultivation experiments[28,57]. Copiotrophic strains grew to the

highest densities in the medium with the highest nutrient content (1:10 diluted NSY containing 0.3 g complex carbon sources per litre), and several of these genera have been previously isolated by the filtration acclimatisation method using NSY[53]. Several strains (*Aquidulcibacter, Zwartia, Pernthalerella* gen. nov., *Leadbetterella, Caulobacter, Sphingobium*) were in between the two extremes of oligo- and copiotrophs and might be considered mesotrophs. Notably, these strains grew also best in the medium with the highest nutrient content, however, maximum abundances were one order of magnitude lower than for copiotrophs while being significantly higher than for oligotrophs (Fig. 2b and Supplementary Data 4, 5).

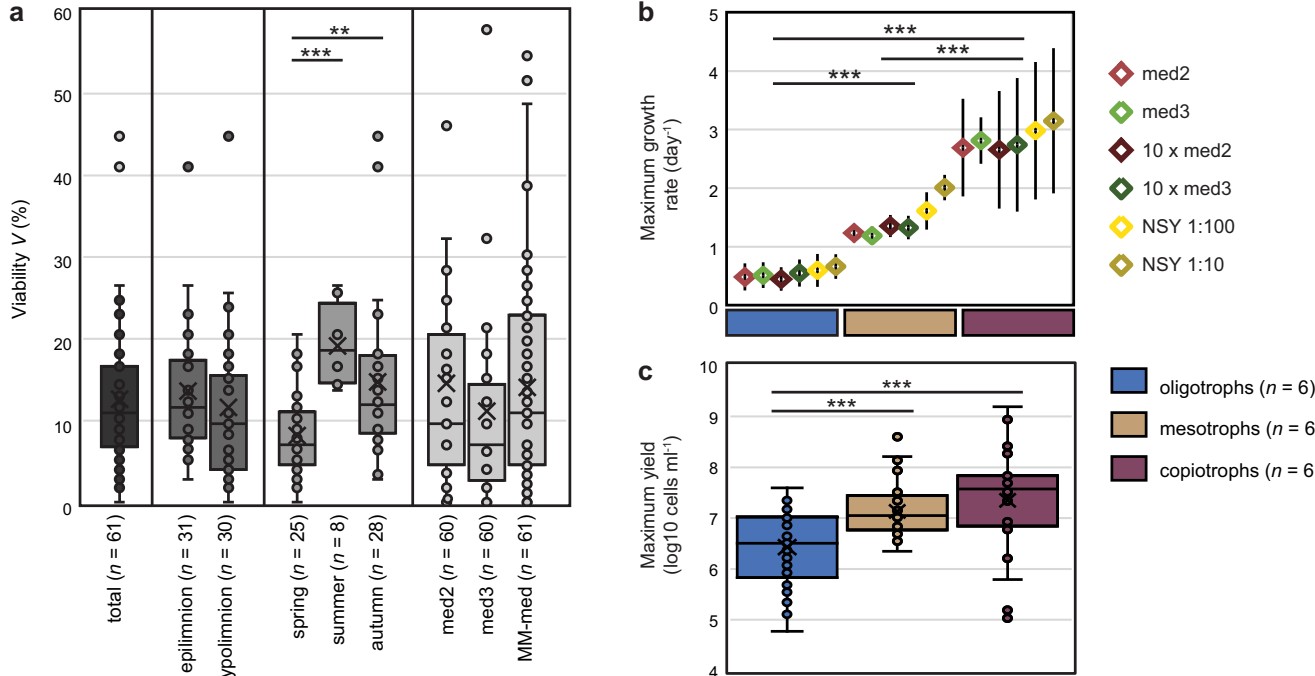

**Fig. 2 | Isolation success and growth characteristics of new strains. a** Viability of dilution-to-extinction cultivation experiments shown separately for all samples (total), samples gained from different depth layers (epi- and hypolimnion), different seasons, and three different cultivation media. The number of samples are given in brackets, significant differences ($t$ tests) are indicated by asterisks (**: $p < 0.01$, ***: $p < 0.001$). Boxes indicate the 25th and 75th quantiles, medians are displayed by central lines, whiskers indicate the 5th and 95th quantiles, and individual values are displayed by open circles. **b** Maximum growth rates of 18 strains grown in six different media with low to high carbon and nutrient content. Shown are averages and standard deviation of maximum growth rates for oligotrophs (blue; $n = 6$ strains), mesotrophs (brown; $n = 6$) and copiotrophs (purple; $n = 6$), that were clustered according to growth characteristics. Significant differences between these groups ($t$ tests) are indicated by asterisks (***: $p < 0.001$). **c** Maximum abundances of 18 strains grown in six different media with low to high carbon and nutrient content. Strains were grouped in oligo-, meso- and copiotrophs according to growth characteristics as in (**b**). Boxes indicate the 25th and 75th quantiles, medians are displayed by central lines, whiskers indicate the 5th and 95th quantiles, and individual samples are displayed by open circles. Significant differences between these groups ($t$ tests) are indicated by asterisks (***: $p < 0.001$). Individual plots and growth curves of all strains are shown in Supplementary Figs. 4 and 5 and clustering of growth rates in Supplementary Fig. 6. Raw data are provided in Supplementary Data 3 and 5, and results of statistical tests in Supplementary Data 4.

## Culture collection includes the most abundant taxa of freshwater lakes

We sequenced metagenomes of the free-living microbial fraction (5–0.22 µm) of the same water samples used for cultivation ($n = 67$) to determine the community composition by taxonomic assignment of 59 single-copy marker genes[58]. The most abundant microbes in metagenomes were affiliated with the phyla *Actinomycetota, Pseudomonadota, Bacteroidota, Verrucomicrobiota, Planctomycetota, Chloroflexota, Cyanobacteriota, Thermoproteota*, and *Patescibacteria* while other phyla contributed on average < 1% (Supplementary Fig. 7 and Supplementary Data 6, 7). Relative abundances based on taxonomic classification using 59 single-copy marker genes were significantly correlated with results obtained by using 16S rRNA genes as taxonomic markers ($R^2 = 0.9725$, $p < 0.0001$, Supplementary Notes, Supplementary Figs. 8–10, Supplementary Data 4, 6, 8). Although a total of 1734 genera were detected based on the 59 single-copy markers, individual samples contained, on average, 248 genera, of which 157 were present in high numbers (>1% in any of the metagenomes). These 157 abundant genera made up the majority of all prokaryotes (on average 84% of all reads, range 64–96%), while the remaining genera were rare and occurred only sporadically (Fig. 3a, Supplementary Fig. 11 and Supplementary Data 7). Most of these abundant genera have been long recognised as ubiquitous freshwater microbes according to 16S rRNA gene amplicon studies and metagenomic surveys[11,59]. Only 39 of the 157 abundant genera have been validly published under the International Code of Nomenclature of Prokaryotes (ICNP) according to the List of Prokaryotic names with Standing in Nomenclature (LPSN[3]), and another

seven were proposed as *Candidatus* genera (Supplementary Data 7). We assigned the GTDB taxonomy to axenic cultures to identify their presence in the original samples (Supplementary Data 3). Our culture collection contained 30 of the 157 abundant freshwater genera, most notably 15 genera within the top 30 most abundant and prevalent genera (Fig. 3a). Only 16 of the 30 abundant genera from our culture collection have been validly described to date[3]. Another 30 genera from our culture collection belonged to the rare biosphere (< 1% in any metagenome), three more were below the detection limit in metagenomes and are either very rare or contaminants, and the rest could not be confidently assigned to GTDB taxonomy. Considering the total diversity, up to 48% of genera present in individual samples (average 15%) were represented by at least one strain from our culture collection. We further evaluated abundance profiles of genera from the culture collection in the initial samples. If translated to relative abundances, we were successful in obtaining isolates that accounted for up to 72.3% of all prokaryotes present in the epilimnion (average 51.61%) and 47.7% in the hypolimnion (average 27.8%) of the sampled lakes (Fig. 3b, c).

Next, we screened 462 public freshwater metagenomes from seven different continents to assess if genera from our strain collection are globally distributed (Fig. 3c and Supplementary Data 10). Indeed, these genera represented on average 16% of the total diversity in individual samples, again with a higher prevalence in the epilimnion than the hypolimnion of lakes. These values correspond to relative abundances of up to 76.5% of all prokaryotes in the epilimnion (average 37.5%) and 42.7% in the hypolimnion (average 20.1%). Slight biogeographic differences were recorded with lower values for Africa

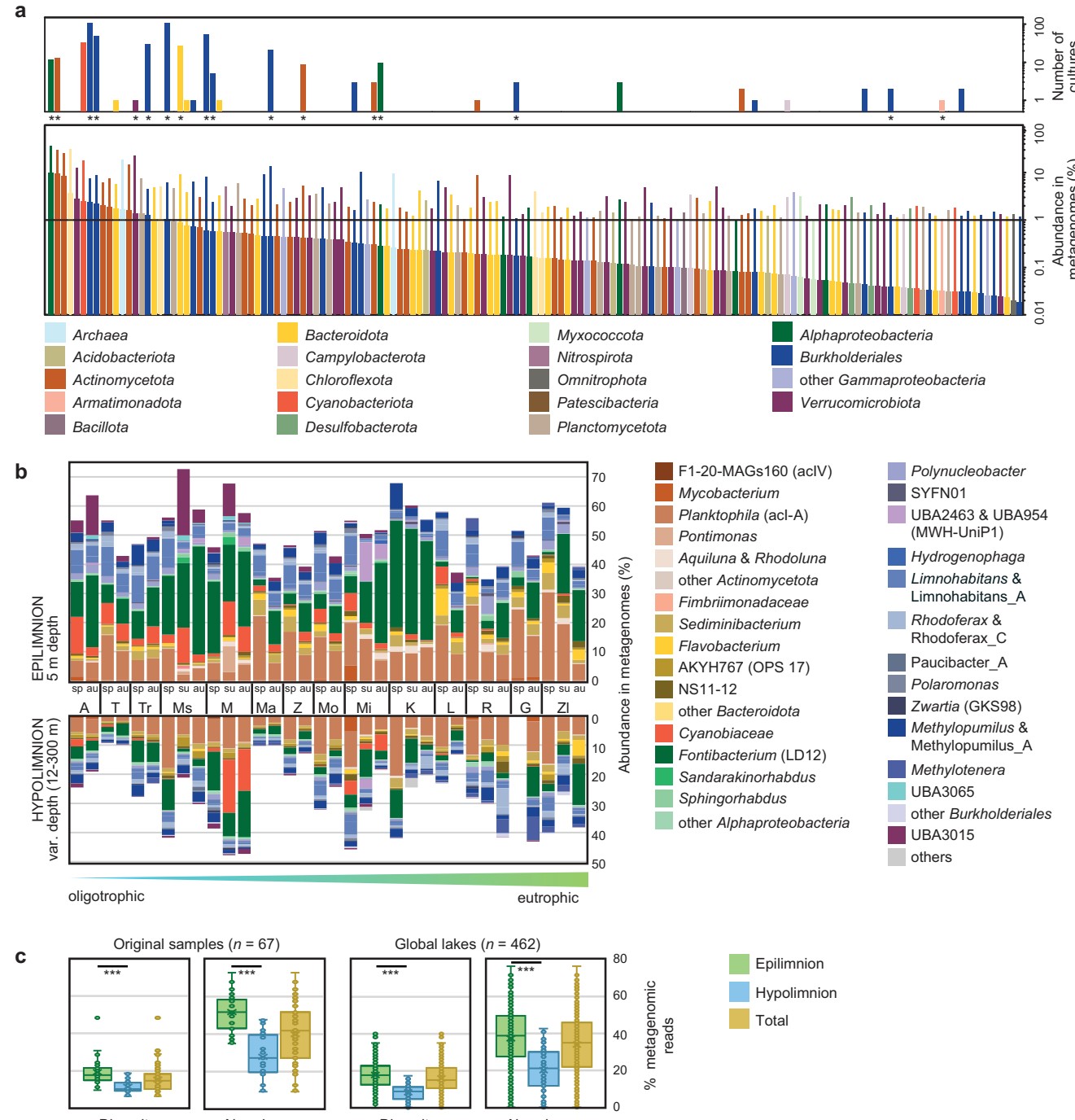

**Fig. 3 | Representation of the culture collection in lake samples. a** Rank-abundance curve (lower panel, averages are shown as bars, maxima as thin lines) and number of axenic cultures (upper panel) of abundant genera present in the sampled lakes (>1% of reads in at least one sample). Metagenomic reads were taxonomy-assigned with SingleM. Asterisks below cultures indicate that at least one member of the genus was genome-sequenced. See Supplementary Fig. 11 and Supplementary Data 7 for all genera, including rare taxa. **b** Summed up relative abundances of taxa with representatives in the culture collection in the epi- and hypolimnion of the sampled lakes. Samples are sorted from oligo- to eutrophic, abbreviations of lakes and sampling seasons are as in Supplementary Fig. 1. **c** Proportion of genera from the culture collection relative to the total number of genera (diversity) and their summed up relative abundances (abundance) in the sampled lakes (original samples, $n = 67$; left panels) and in 462 publicly available metagenomes from seven continents (global lakes, $n = 462$; right panels). Metagenomic reads were taxonomy-assigned with SingleM, proportions of genera included in the culture collection are separately shown for different water layers (epilimnion, green; hypolimnion, blue) and total (ochre). Boxes indicate the 25th and 75th quantiles, medians are displayed by central lines, whiskers indicate the 5th and 95th quantiles, individual samples are displayed by open circles. Significant differences between epi- and hypolimnetic samples (t tests) are indicated by asterisks (***: $p < 0.001$). Raw data are provided in Supplementary Data 7 and 10, results of statistical tests in Supplementary Data 4.

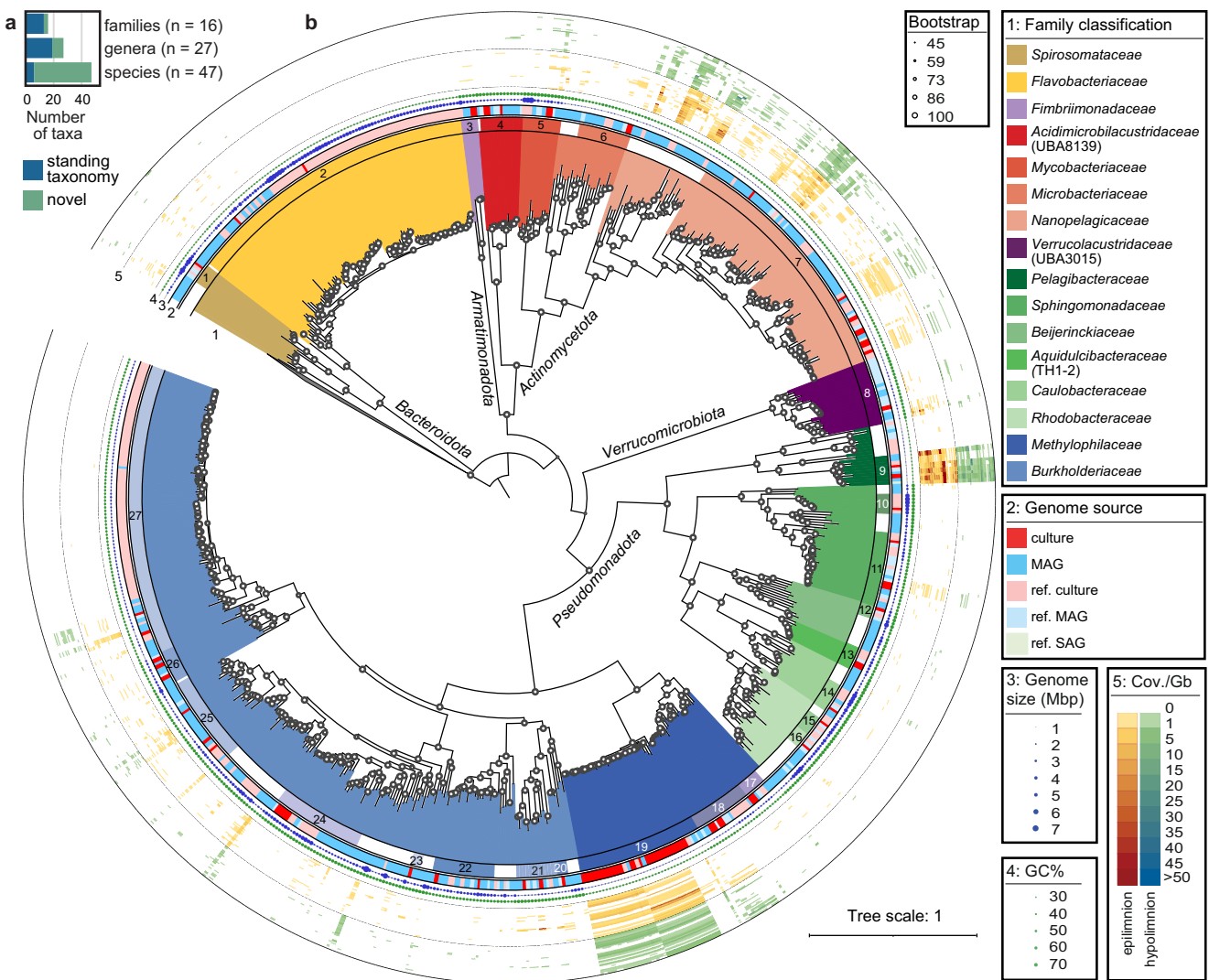

**Fig. 4 | Diversity and phylogeny of genome-sequenced cultures compared to metagenome-assembled genomes (MAGs) assembled from the same samples.** **a** Number of genome-sequenced cultures with standing taxonomy (blue) and novel taxa (turquoise) on different taxonomic levels (family, genus, species). **b** Phylogenomic tree of 120 single copy marker protein sequences of bacterial families containing cultures (red, *n* = 87), MAGs assembled from the same water samples (blue, *n* = 313), and closely related published genomes (*n* = 324) from cultures (light red, ref. culture), MAGs (light blue, ref. MAG), and SAGs (light green, ref. SAG). Five genomes of *Patescibacteria* were used to root the tree. Clades are coloured on a family level and genera containing cultures are marked in different colours and numbers (1: *Leadbetterella*; 2: *Flavobacterium*; 3: *Fimbriicoccus* gen. nov.; 4: *Acidimicrobilacustris* gen. nov.; 5: *Mycobacterium*; 6: *Rhodoluna*; 7:

*Planktophila*; 8: *Verrucolacustris* gen. nov.; 9: *Fontibacterium*; 10: *Sphingobium*; 11: *Sphingorhabdus*; 12: *Rhabdaerophilum*; 13: *Aquidulcibacter*; 14: *Caulobacter*; 15: *Tabrizicola*; 16: *Allotabrizicola* gen. nov.; 17: *Novimethylotenera* gen. nov.; 18: *Methylotenera*; 19: *Methylopumilus*; 20: *Lacustribacter* gen. nov.; 21: *Hahnella* gen. nov.; 22: *Zwartia*; 23: *Hydrogenophaga*; 24: *Allorhodoferax* gen. nov.; 25: *Limnohabitans*; 26: *Pernthalerella* gen. nov.; 27: *Polynucleobacter*). Annotations of the rings from inside to outside: Genome source; genome size (blue bubbles); genomic GC content (green bubbles); metagenomic fragment recruitment for 67 samples from where cultures and MAGs were obtained. The lakes are sorted from oligo- to eutrophic as in Fig. 3, and epi- and hypolimnetic samples are shown separately in shades of orange-red and green-blue, respectively. See Supplementary Fig. 14 for a full tree with all 1294 dereplicated MAGs covering all families.

(17.3%, *n* = 44) and Australia (18%, *n* = 9) that could partly be explained by a large number of anoxic samples from African Rift Valley lakes or global biogeographic distribution patterns (Supplementary Figs. 12, 13 and Supplementary Data 4, 9)[60,61]. Nevertheless, these results confirm the ubiquitous distribution and global relevance of genera from our culture collection.

We further evaluated the species-level abundance by metagenomic fragment recruitment. Eighty-seven genome-sequenced strains from our culture collection (Supplementary Data 11), MAGs assembled from the same water samples (Supplementary Data 12), and 324 genomes of their closest relatives obtained from public databases (Supplementary Data 13) served as a reference for metagenomic read mapping (Fig. 4b, Supplementary Fig. 15, Supplementary Data 16 and Supplementary Notes). Five species of the genus *Fontibacterium*

(including the two new species isolated here) were by far the most abundant prokaryotes in the 67 original samples (up to 117 × coverage per Gb mapped data). Other abundant strains from our culture collection (*Hahnella* gen. nov., *Methylopumilus, Planktophila, Methylotenera, Lacustribacter* gen. nov., *Mycobacterium, Allosphingorhabdus* gen. nov., *Hydrogenophaga, Verrucolacustris* gen. nov., *Rhodoluna, Limnohabitans, Fimbriicoccus* gen. nov.) were present at max. 2-39 × coverage. In addition, we mapped 250 publicly available metagenomes from six continents and two seasonally resolved time-series (Lake Mendota, USA, *n* = 94[62], Řimov Reservoir, Czechia, *n* = 81[52,63]) to our genome collection (Supplementary Notes, Supplementary Figs. 15, 16 and Supplementary Data 17). Most species were present in multiple metagenomes, confirming their global presence and generally high relevance in freshwater habitats.

## Culture collection is taxonomically novel and metabolically diverse

We selected 87 strains from our culture collection for whole-genome sequencing, including many methylotrophs (*Methylopumilus*, *Methylotenera*, *Novimethylotenera* gen. nov., n = 36[50,52];) that were enriched in a medium containing methanol and methylamine as sole carbon sources. Most genomes could be curated to a circular chromosome (n = 63, *Rhabdaerophilum aquaticum* MsE-M23 sp. nov. also contained a circular plasmid), and the remaining genomes were in 1–44 contigs (Supplementary Data 11). GTDB classification[5] and ANI and AAI comparisons (Supplementary Data 14, 15)[64] indicated that the genomes represent 27 genera and 47 distinct species by using AAI cutoffs of 65% and ANI cutoffs of 95%, respectively. Only one genome could be assigned to a validly described species (*Sphingobium cupriresistens*[65]), and five more were previously proposed as *Candidatus* species by us[24,50] (Supplementary Fig. 17). Moreover, genome-sequenced cultures contained nine novel genera and two novel families (Fig. 4a and Supplementary Data 11). We putatively named these new taxa according to SeqCode[66] as follows: *Acidimicrobilacustris* gen. nov. (*Acidimicrobilacustridaceae*, fam. nov., *Actinomycetota*), *Verrucolacustris* gen. nov. (*Verrucolacustridaceae*, fam. nov.; *Verucomicrobiota*), *Fimbriicoccus* gen. nov. (*Armatimonadota*), *Allotabrizicola* gen. nov., *Hahnella* gen. nov., *Lacustribacter* gen. nov., *Novimethylotenera* gen. nov., *Allorhodoferax* gen. nov., and *Pernthalerella* gen. nov. (*Pseudomonadota*; see Supplementary Notes for detailed description and etymology).

We compared the 87 genomes from cultures to 1294 medium-high quality MAGs (> 50% completeness, < 5% contamination) obtained from the same water samples and dereplicated at the species level (95% identity) (Supplementary Data 12) and to 324 genomes of their closest relatives obtained from public databases (Supplementary Data 13). Only seven MAGs could be assigned to a described species and another seven to *Candidatus* species (1.1% of the 1294 MAGs), reiterating the high proportion of undescribed taxa in freshwater environments[4,11]. In the majority of cases, strains from our culture collection were phylogenetically closer to MAGs than to previously cultured species, and only one, *Sphingobium cupriresistens*, is validly described and available in a culture collection[65] (Fig. 4b and Supplementary Figs. 14, 17), exemplifying that the genome-sequenced cultures were indeed novel and highly relevant.

Genomes were further assessed for characteristics indicative of genome streamlining and metabolic pathways relevant in freshwaters[11,14]. Genome sizes ranged from 1.06–4.99 Mbp with GC contents of 29.4-67.6% (Fig. 5, Supplementary Fig. 18 and Supplementary Data 10). We observed clear indications for genome streamlining in cultures with genome sizes < 1.5 Mbp (*Fontibacterium*, *Methylopumilus*, *Planktophila*, *Rhodoluna*), and significant correlations between genome size and coding density, genomic GC content (except for *Bacteroidota* that are known to have a low GC content), intergenic spacers, numbers of tRNAs, paralogs, sigma factors, and signal transduction genes (Supplementary Fig. 18 and Supplementary Data 4) as previously reported[11,14,24,50]. Genome-streamlined strains tended to use a different stop codon (TAA), indicative of nitrogen limitation[50], which is also reflected in a low GC content, and contained more membrane transporters (normalised per Mbp genome size) than strains with medium to large genomes. Genome-streamlined bacteria were also clearly more abundant than microbes with larger genomes, both in the original water samples (n = 67) and in freshwater metagenomes gained from six continents (n = 250). Further, microbes containing genes encoding motility and secretion systems type II, IV, or VI had significantly larger genome sizes (Supplementary Fig. 18 and Supplementary Data 4, 18), except for two *Zwartia* strains with large genomes (3.6 and 4.5 Mbp) that lacked genes for flagella assembly and chemotaxis, congruent with previously described *Zwartia* strains[67]. The phage-defence system CRISPR-Cas[68,69] was present in only 5 strains with medium-large genome sizes (2.6-5

Mbp, Supplementary Fig. 19 and Supplementary Data 19). In contrast, 20.8% of the reference genomes obtained from cultures and 13.9% of MAGs contained on average 1.7 CRISPR arrays per genome. There was a clear relationship to genome size, as microbes containing CRISPR-Cas systems were significantly larger than those without (p = 7.524E-47, Supplementary Data 4).

Most genome-sequenced cultures can potentially utilise light as an energy source, either via proton-pumping rhodopsins (n = 59) or by being anoxygenic aerobic phototrophs (AAPs, n = 15). Only 13 strains contained neither mechanism for harvesting light energy (Fig. 5 and Supplementary Data 18). This reiterates the importance of light-induced energy conservation in aquatic bacteria[70,71], with rhodopsin-bearing microbes being significantly smaller in genome size than AAPs (Supplementary Fig. 18). Interestingly, five AAPs additionally contained a carbon fixation pathway (Calvin cycle via RuBisCO type I), potentially leading to photoautotrophy that was so far only occasionally reported for sediment-dwelling AAPs[72]. Indeed, one strain (*L. simekii* MiE-M12, sp. nov.) was able to grow to densities of $5 \times 10^6$ cells ml$^{-1}$ with a growth rate of 1.2 d$^{-1}$ in a medium without any organic carbon compounds except for vitamins (Supplementary Fig. 20). While growth was observed in both dark and light:dark (12:12 h) conditions, *L. simekii* maintained significantly higher cell numbers during stationary phase in light:dark conditions (p = 0.009175, Supplementary Data 4) hinting at photoautotrophic growth during starvation. Pathways related to the utilisation of different nitrogen and sulphur compounds were variable and mainly present in cultures with medium to large genome sizes. Further, we observed an occasional presence of degradation pathways for aromatic compounds and a widespread ability of glycolate oxidation in *Pseudomonadota*. Methylotrophy via canonical methanol oxidation and the RuMP cycle was restricted to *Methylophilaceae,* with *Methylotenera* and *Novimethylotenera* gen. nov. also containing the methylcitric acid (MCA) cycle[50] (Fig. 5, Supplementary Data 20). Several genomes encoded the potential for the utilisation of other C1 compounds such as methylamine (*Pernthalerella* gen. nov.), dimethylamine/trimethylamine or dimethylsulfoniopropionate (DMSP) oxidation (*Tabrizicola*, *Allotabrizibola* gen. nov.). On the other hand, no indications for C1 metabolism were detected in the genome of *Verrucolacustris* gen. nov., although all described relatives are methanotrophs (*Methylacidiphilales*)[73]. Carbohydrate active enzymes (CAZy) were present in variable numbers in genome-sequenced strains, with the highest occurrence in *Bacteroidota* and the novel *Armatimonadota* genus *Fimbriicoccus* gen. nov. and lowest in genome streamlined strains (Fig. 5 and Supplementary Data 21). ABC transporters showed an opposite trend; < 1 transporter per Mbp genome size were identified in *Bacteroidota*, while up to 10 transporters per Mbp were present in *Rhodoluna*, *Planktophila*, *Hydrogenophaga* and *Allorhodoferax* gen. nov. (Fig. 5 and Supplementary Data 18). Some genera (*Allorhodoferax* gen. nov., *Planktophila*, *Hydrogenophaga*, *Allotabrizicola* gen. nov.) had a higher proportion of carbohydrate transporters, while others (*Fontibacterium*, *Rhodoluna*, the majority of *Burkholderiaceae*) seem to rely more on amino acid transporters. Most strains were prototrophs for amino acids (ranging from 16 to 20 amino acids), while many genomes lacked biosynthesis pathways for vitamins (ranging from prototrophy for only three vitamins to a maximum of seven vitamins). Auxotrophy for cobalamin was most common, as reported previously for marine and freshwater microbes[74], however, our culture collection also contains prototrophs (*Mycobacterium*, *Tabrizicola*, *Allorhodoferax* gen. nov.), that might provide cobalamin as a public good to co-occurring microbes in the plankton of lakes[15,75]. Heme auxotrophy was observed for *Planktophila*, *Rhodoluna*, *Fimbriicoccus* gen. nov. and *Verrucolacustris* gen. nov[17]. In general, many metabolic features were not conserved on the genus level, such as the presence of rhodopsins (*Zwartia*, *Methylotenera*), the ability for carbon fixation (*Pernthalerella* gen. nov., *Hahnella* gen. nov., *Limnohabitans*), nitrate reduction (*Zwartia*, *Allorhodoferax* gen. nov.), methanesulfonate oxidation

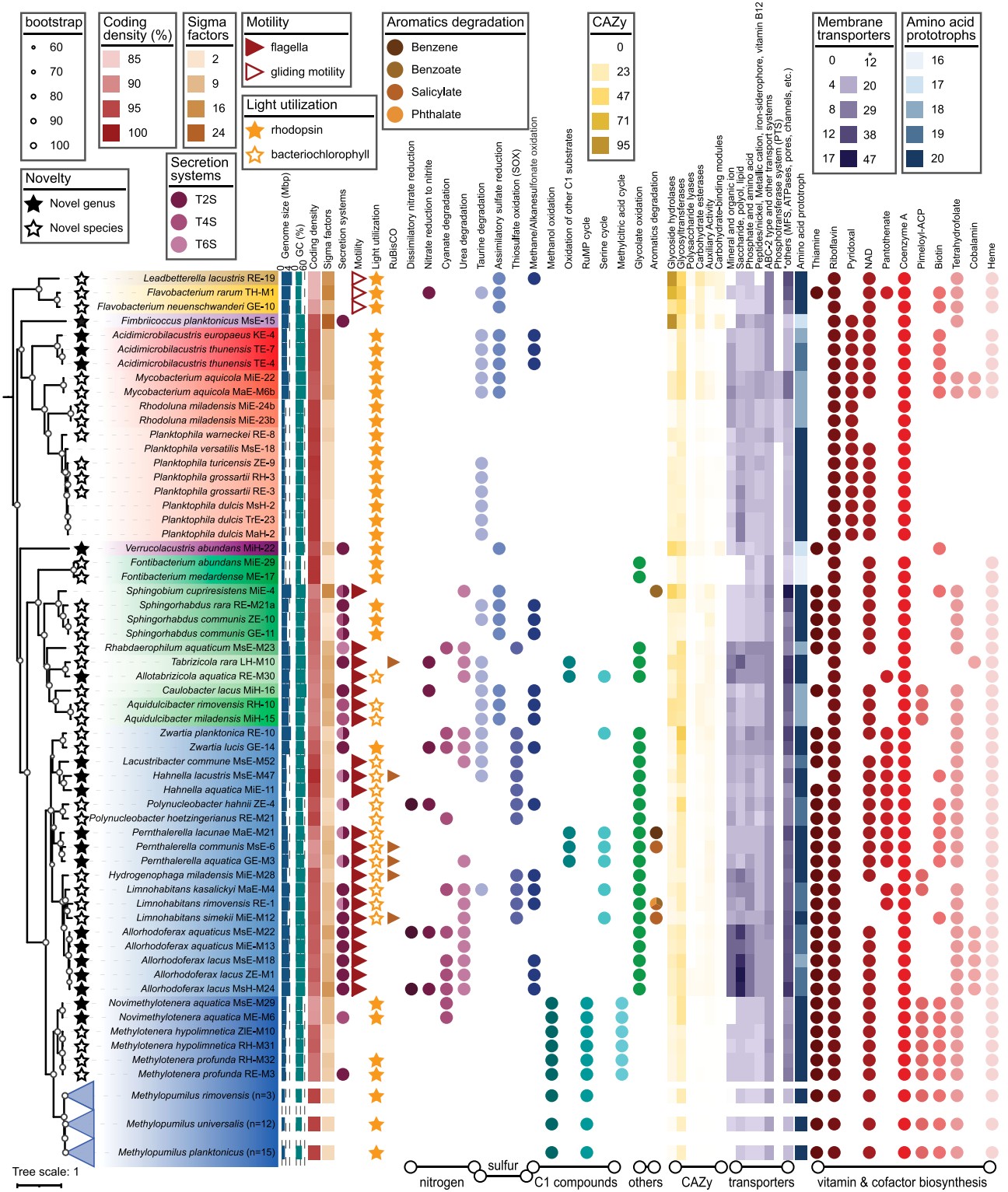

**Fig. 5 | Phylogeny, genomic characteristics, and selected metabolic pathways of 87 genome-sequenced strains of the culture collection.** The bootstrapped phylogenomic tree was constructed with 120 single-copy marker genes. Novel taxa are indicated by open (species) and filled (genera) stars. Details on metabolic pathways can be found in Supplementary Data 11, 18, 19, and 20. Please note that general secretion systems Sec and Tat are not shown here, as these were present in all strains.

(*Zwartia*), degradation of aromatic compounds (*Pernthalerella* gen. nov., *Limnohabitans*), or vitamin biosynthesis. Some of these pathways are known to be variable with a high frequency of horizontal gene transfer between co-occurring microbes[24,76]. Although we genome-sequenced only one-sixth of the culture collection, the diversity of metabolic pathways and life strategies was surprisingly high.

## Future directions

Our cultivation strategy showcases the possibility to obtain a large diversity of taxa by using appropriate methods in a high-throughput manner. Dilution-to-extinction cultivation with artificial media mimicking natural oligotrophic freshwater conditions proved to be a powerful method to obtain highly abundant strains that were so far recalcitrant to cultivation with traditional techniques. We named 41 taxa according to SeqCode[66], including higher-level taxonomic ranks (nine genera, two families), thus reducing the number of complicated alphanumerical placeholder names used in genomic studies[5]. Our culture collection did not only provide complete genomes for detailed evolutionary and ecogenomic analyses (e.g., microdiversity[52,77]) that can serve as reference in future meta-omic surveys, the strains can also be used to test hypotheses derived from culture-independent studies[17,49,78]. The establishment of novel model organisms is highly needed, as most currently used model taxa are copiotrophs that fundamentally differ from environmentally abundant oligotrophs[79]. Metabolic and physiological characterisation of selected model strains, experiments targeting interactions with other organisms (e.g., competition and cooperation, predation by protists and viruses[57,80,81]) and responses and adaptations to changing environmental conditions (e.g., via transcriptomics, proteomics or metabolomics[82,83]) will foster a deeper understanding of the ecology of abundant freshwater microbes.

## Methods

### Sampling of 14 lakes in Central Europe

Fourteen lakes located in Austria (Mondsee, Attersee, Traunsee), Czechia (Medard, Most, Milada, Římov Reservoir, Klíčava Reservoir, Žlutice Reservoir), Italy (Maggiore, Lugano) and Switzerland (Greifensee, Thun, Zurich) were sampled in spring and autumn 2019, and four Czech lakes (Medard, Most, Milada, Římov Reservoir) additionally in summer 2019 (Supplementary Fig. 1 and Supplementary Data 2). These lakes are of different trophic states (ultraoligotrophic to eutrophic), sizes (0.6–212 km²) and max. depths (17–370 m). Vertical profiles of water temperature, conductivity, pH, oxygen, and chlorophyll *a* concentrations were recorded with a submersible probe (YSI EXO3, Yellow Springs Instruments, Yellow Springs, USA). Two different water layers each were sampled with a Niskin bottle or Friedinger sampler for the isolation of microbes and metagenomics (Fig. 1): the upper water layer (epilimnion, 5 m depth) and the oxygenated deep-water layer (hypolimnion, variable depth depending on the maximal depth, Supplementary Data 2). A total of 3-34 l of each water sample was prefiltered via 20 μm plankton nets and serially filtered through 5 μm (Sterlitech PES membrane filters, USA) and 0.22 μm (Millipore Express PLUS, Germany) polysulfone filters with a peristaltic pump until filters were clogged, and filters were stored at −80 °C until DNA isolation. Two hundred ml of the 0.22 μm filtrate were collected for nutrient analyses (phosphorus, nitrate and dissolved organic carbon)[84].

### High-throughput dilution-to-extinction cultivation

Ten ml water samples from each depth were filtered through 0.4 μm filters to remove larger organisms. Prokaryotic abundances were quantified with a CytoFLEX S flow cytometer (Beckman Coulter; Brea, CA, USA) equipped with a blue laser (488 nm, bandpass filters 525/40 and 690/50) after staining with SYBRgreen I (0.5 × final concentration; Lonza, Rockland, ME, USA) and analysed with CytoExpert version 2.4.0.28. Samples were diluted and inoculated (approx. 1 cell per well; range 0.5 - 5 cells per well) to 96-deep-well plates filled with two different media based on ALW (artificial lake water)[24,85] containing sodium bicarbonate (2 mM), potassium nitrate (30 μM), magnesium sulphate (200 μM), calcium chloride (40 μM), calcium sulphate (40 μM), ammonium chloride (30 μM), the trace element solution SL-4 (https://mediadive.dsmz.de/solutions/20) and low concentrations of

diverse carbon and nutrient sources (Supplementary Data 1). One medium (med2) contained 0.5 μM each of D-glucose, D-ribose, sodium pyruvate, sodium citrate, oxaloacetic acid, sodium acetate, sodium succinate, glycerol, urea, taurine, ethanol, and sodium thiosulfate, and 0.1 μM of N-acetylglucosamine, adding up to 1.13 mg dissolved organic carbon (DOC). The second medium contained the same components and additionally 0.5 μM each of putrescine, spermidine, D-xylose, D-arabinose and glycolate, adding up to 1.33 mg DOC. After autoclaving, both media were amended with a filter-sterilized mix of the 20 proteinogenic amino acids (0.2 μM of each amino acid except for glutamate and glutamine which were added at 0.4 μM concentration), vitamins (0.593 μM thiamine, 0.08 μM niacin, 0.074 nM cobalamin, 0.005 μM para-amino benzoic acid, 0.074 μM pyridoxine, 0.081 μM pantothenic acid, 0.004 μM biotin, 0.004 μM folic acid, 0.555 μM myo-inositol), dipotassium phosphate (3.22 μM) and catalase (10 U ml⁻¹)[86]. In addition, 1 plate each was prepared with a medium (MM-med) containing the C1 compounds methanol (1 mM) and methylamine (0.1 mM) as sole carbon source to select for methylotrophs[50,87]. Methylotrophs were enriched in MM-med (1:1 filtered sample to medium) at 16 °C for 24–48 h prior to inoculation in the MM-med plate, while plates with all other media were inoculated as soon as possible (within 1–5 h)[50,52,87]. A total of 64 96-well plates were incubated in a climate-controlled room (16 °C) in 12:12 h light:dark cycle for 6–8 weeks (Fig. 1 and Supplementary Data 3). Growth in individual wells was checked by flow cytometry as described above, examples of wells with and without growth are shown in Supplementary Fig. 21. All wells with microbial growth (> 10⁴ cells ml⁻¹) were transferred to fresh medium and further maintained by 10-week cycles consisting of 1:10 transfers (0.5 ml culture to 4.5 ml medium), six week incubation, 1:1 fills (adding 5 ml fresh medium), followed by another 4 week incubation after which cultures were measured on the flow cytometer and again 1:10 transferred to start the next cycle. Glycerol stocks and 16S rRNA gene screening and analyses were performed as previously described[24,50]. Isolation success expressed as Viability (*V*), i.e., probability that a cell selected at random is viable was calculated based on the formula by Button et al.[48] as follows:

$$V = \frac{\ln(1-p)}{X}$$

Where *p* is the number of wells or cultivation tubes, *n*, with growth *z* ($p = z/n$) and *X* is the estimated number of cells inoculated per well.

Strains from the culture collection can be requested by email to Michaela M. Salcher (michaelasalcher@gmail.com).

### Growth characterisation of cultures

Growth of 18 selected cultures was characterised in short-term growth assays in up to eight different media (Supplementary Data 4): med2, med3, MM-med containing methanol and methylamine as sole carbon sources[50], med2 containing additionally 1 mM methanol and 100 nM methylamine (med2 + MM), 10 × med2, 10 × med3 (both containing 10 × concentrations of carbon sources and phosphate), 1:100 diluted NSY containing 0.01 mg each of nutrient broth, soytone and yeast extract, and 1:10 diluted NSY containing 0.1 mg each of nutrient broth, soytone and yeast extract[88]. Cultures were inoculated in triplicates at approx. $1 \times 10^4$ cells ml⁻¹ in 96-deep-well plates containing 1.5 ml of medium and monitored by flow cytometry 2-3 times per week for a period of 17–32 days. Examples of flow cytograms of four different strains are shown in Supplementary Fig. 21. Growth rates and maximum cell yield per medium were calculated as previously described[54,57]. We performed hierarchical clustering to group strains in different growth strategies (oligo-, meso-, copiotrophs). Maximum growth rates in med2, med3, 10 × med2, and 10 × med3 were transformed (log10) to calculate a dissimilarity matrix using Euclidean distance. Hierarchical clustering was then applied to this matrix using

Ward's minimum variance method (hclust function, method = ward.D2) in the R[89] package cluster v2.0.7-1[90]. The optimal number of species clusters was determined by evaluating the dendrogram structure alongside the average silhouette width (cluster::silhouette function).

## Whole-genome sequencing

Eighty-seven cultures were selected for whole-genome sequencing and grown in 500 ml Erlenmeyer flasks until the late stationary phase. Biomass was harvested by centrifugation at $18,549 \times g$ for 60 mins at room temperature or by filtration onto $0.22\,\mu m$ (Millipore Express PLUS, Germany) polysulfone filters, and DNA was isolated using the Qiagen MagAttract HMW DNA kit according to the manufacturer's instructions. In addition, the DNA of one culture with growth arrest (*Verrucolacustris abundans* MiH-22) was amplified with multiple displacement amplification with the REPLI-g single cell kit (Qiagen, Venlo, The Netherlands) as described previously[24]. Paired-end libraries were constructed (PE150) and sequenced on an Illumina NovaSeq 6000 instrument (NOVOGENE, HK). Raw reads were trimmed using BBMap v36.x (https://github.com/BioInfoTools/BBMap/) and assembled with SPAdes v3.12.0 (using k-mers 29,39,49,59,69,79,89,99,109,119,129)[91]. The majority of assemblies resulted in < 5 contigs that could manually be curated to a circular chromosome after repeated rounds of mapping of trimmed reads to contigs with Geneious 10 (default mapper, high sensitivity; www.geneious.com), extending contigs on both ends, identifying overlapping ends and assembling with the Geneious 10 assembler (de novo assembly, high sensitivity). Three *Methylopumilus* genomes were closed by designing primers bordering gaps, PCR amplification and Sanger sequencing of the amplicon[52].

## Metagenomic sequencing, taxonomy assignment with SingleM and 16S rRNA classification

DNA from $0.22\,\mu m$ lake water metagenome filters was extracted using the ZR Soil Microbe DNA MiniPrepTM kit (Zymo Research, Irvine, CA, USA) according to the manufacturer's instructions and paired end libraries (PE150) were sequenced on an Illumina NovaSeq 6000 instrument (NOVOGENE, HK.). Reads were quality controlled (Q > 33), trimmed (trimq = 18 qtrim = rl), and adaptors were removed using BBMap v36.x[92].

We used SingleM[58] for phylum- (Supplementary Data 6) and genus-level (Supplementary Data 7) classification (SingleM commands pipe and summarise) of quality filtered reads based on conserved regions of 59 single copy marker genes and the Genome Taxonomy Database (GTDB r214[5]). In addition, SingleM taxonomic profiles from 462 public metagenomes were retrieved from Sandpiper 0.2.2 (https://sandpiper.qut.edu.au/) and further analysed at the genus-level using SingleM summarise[58]. To determine abundance of our isolate collection in these metagenomes, 16S rRNA gene sequences of the isolate collection were assigned to the SILVA SSU database RefNR99 138.1[93] and taxonomically compared to GTDB as follows (Supplementary Data 3): Matches to genome-sequenced representatives (Supplementary Data 11), matching genus names, MAGs containing 16S rRNA genes (Supplementary Data 12), and literature research. Of the 72 genera 59 could be confidently assigned, while some 16S rRNA gene sequences could only be assigned to a family or order level (*Cyanobiaceae*, NS11-12, AKYH767, *Arcobacteraceae*, *Micropepsaceae*) or had multiple hits to genome-sequenced cultures (Methylotenera_1 = *Methylotenera* and Methylotenera_A; MWH-UniP1 aquatic group = g_UBA2463 and g_UBA954; all four genera were part of our genome-sequenced culture collection). These taxa were summed up in the SingleM matrix. In four cases, we could not determine a corresponding GTDB genus taxonomy from 16S rRNA taxonomy assigned by SILVA (uncultivated *Methylophilaceae*, uncultivated *Frankiales*, *Heliimonas* and *Rhodobacter*). These four taxa were removed from further analyses. Further, the SILVA taxonomy did not match genus names of several genome-sequenced strains or MAGs (uncultured

*Comamonadaceae* = *Limnohabitans*, *Limnohabitans* = Limnohabitans_A, *Comamonadaceae* = Rhodoferax_C, *Asinibacterium* = *Sediminibacter*) and was changed accordingly.

In addition to SingleM analyses, 16S rRNA gene classification of metagenomes was performed. Metagenomes were subsampled to 20 million reads, and 16S rRNA genes were predicted by using ublast and SSU-ALIGN[94] with the RDP release 11 database[95] clustered at 90% identity. The SILVA SSU database RefNR99 138.1 was used for taxonomic assignment[93]. Hits were normalised by rRNA copy number following the rrnDB[96] and manual curation (i.e., taxa with different naming in SILVA and rrnDB databases, uncultivated taxa for which we sequenced the first representatives). In case of unknown taxa in the rrnDB, the next higher taxonomic category was used for rRNA copy number estimation (Supplementary Data 9). The results of both methods (SingleM and 16S rRNA gene classification) were significantly correlated (Supplementary Fig. 8). According to the relative proportion of individual taxonomic categories in individual metagenomes, all genera/lineages were classified as either abundant (> 1% of reads in any dataset) or rare (< 1% of reads).

## Metagenomic assembly and binning to MAGs (metagenome assembled genomes)

De novo assembly of curated Illumina reads was done with MEGAHIT v1.1.4-2[97] using default parameters and the following k-mers: 29,49,69,89,109,119,129,149. Four samples were additionally sequenced using the Nanopore PromethION platform (Supplementary Data 2). For these, quality-controlled and trimmed Illumina short-read sequences were used for polishing noisy long-read sequences by generating a Burrows Wheeler Transform (BWT), according to the ropebwt2 construction approach[98]. Nanopore basecalled long-reads with a Q score ≥ 8 were subjected to adaptor and barcode trimming by Porechop[99] and further polished using the generated Illumina BWT with FMLRC2 v0.1.8 with default parameters[100]. Polished long-reads were assembled using Flye v2.9.1-b1780 (settings: --nano-corr −meta −no-alt-contigs)[101]. For both Illumina and long-read assemblies, only contigs ≥ 3 kbp were further used for hybrid binning (tetranucleotide frequencies and coverage data) using MetaBAT2 with default parameters[102]. Bins were manually curated[103] and CheckM2 v1.0.1[104] was used to estimate bin completeness, contamination, and strain heterogeneity. Bins with ≥ 50% completeness and < 5% contamination were selected for further analysis (4342 bins) and dereplicated using dRep (average nucleotide identity (ANI) > 95%[105]), resulting in 1294 representative MAGs (metagenome assembled genomes with the highest dRep score) (Supplementary Data 12). All MAGs and genomes from cultures were classified with the GTDB-Tk v2.4.0 toolkit[106] based on the GTDB releases r214 and r220.

## Phylogenomic and metabolic analyses

A total of 87 genomes from cultures were further used for phylogenomic analyses together with 1294 dereplicated MAGs (Supplementary Data 12) and closely related reference genomes (Supplementary Data 13). Taxonomic classification was done with GTDB r220 as outlined above, and the closest relatives of each culture genome ($n = 324$) were selected in GTDB by verbatim name matching of the assigned genus or family name and literature research and downloaded from NCBI. All genomes from cultures were used to compute average nucleotide (ANI) and average amino acid identities (AAI)[64], including the closest relatives to delineate species and genera at 95% ANI and 65% AAI, respectively (Supplementary Data 14 and 15). Prokka 1.11[107] was used for gene prediction, and annotation was done with hmmsearch[108] against InterProScan v5.46-81.0[109], COG[110], TIGRFAM v15.0[111] and KEGG databases[112]. CRISPR-Cas systems were predicted with PILER-CR v1.06[113]. Carbohydrate-active enzymes (CAZy) were predicted using hmmscan[108] and the dbCAN CAZyme domain HMM database v9 (release date August 2022[114]). Methylotrophic pathways were predicted as described before[50].

A phylogenomic tree of all 87 genomes from cultures and 1294 de-replicated MAGs was calculated with GTDB-Tk[106] using FastTree[115] and pplacer[116]. A detailed tree including all bacterial families with cultures was constructed by using 120 conserved proteins[117], of which at least 46 had to be present in the individual genomes (87 cultures, 313 MAGs and 324 closely related reference genomes). Five genomes of *Patescibacteria* were used as an outgroup to root the tree. Protein sequences were aligned with PRANK[118], trimmed with BMGE v1.12 (-m BLOSUM30 -t AA -g 0.5 -b 3)[119], concatenated (https://github.com/nylander/catfasta2phyml), and a maximum likelihood tree was constructed with IQ-TREE v2.1.2[120] with ultrafast bootstrapping (1000 bootstrap replicates) and the best-fit model LG + F + I + G4, chosen by ModelFinder[121]. Finally, a tree containing 87 culture genomes was constructed the same way (at least 85 markers present in individual genomes, model LG + F + I + G4).

The description of two novel families, nine genera and 41 species according to SeqCode[66] is available as Supporting Note.

### Metagenomic fragment recruitment

All genomes from cultures ($n = 87$), dereplicated MAGs ($n = 1294$) and closely related references ($n = 324$) were used for metagenomic fragment recruitment from 67 metagenomes that were sequenced from the same water samples used for cultivation campaigns (Supplementary Data 2). rRNA genes in genomes were masked prior to recruitment. MMseqs2[122] was used to map metagenomes (subsampled to 20 million reads) to each individual genome to obtain base coverage per gb (-minid 0.95 -mincov 0.9 -minlen 50). In addition, we used 250 publicly available metagenomes from lakes from six continents and two seasonally resolved metagenomic multiannual time series from Lake Mendota, USA (2008–2012, $n = 94$[62]), and Řimov Reservoir, Czechia (2015–2019, $n = 81$[37,52,63]), for fragment recruitment using 47 culture genomes dereplicated at the species level (95% ANI, Supplementary Data 17).

### Reporting summary

Further information on research design is available in the Nature Portfolio Reporting Summary linked to this article.

## Data availability

The sequence data generated in this study have been deposited in ENA under BioProject accession numbers PRJEB77526 (culture genomes) and PRJEB35640 (MAGs and raw metagenomic reads). Accession numbers of metagenomes are listed in Supplementary Data 2, accession numbers of culture genomes in Supplementary Data 11, and accession numbers of MAGs in Supplementary Data 12. Novel families, genera, and species were registered at SeqCode[66] in register lists seqco.de/r:o5wsiwof and seqco.de/r:opjv7zsc. Phylogenomic trees are available at iTOL (https://itol.embl.de/shared/eik87zth8PG1). Source data are provided in this paper.

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

## Acknowledgements

We thank E. Loher, J. Pernthaler, A. Kust, S. Mayer, P. Pejsar, and P. Rychtecký for help during sampling and R. Mala and F. Kostanjšek for excellent laboratory support. The team of K. Řeháková is acknowledged for help in sampling of lakes Medard, Most and Milada and the Laboratory for Water and Soil Protection of the Canton of Bern (GBL), Switzerland, for help in sampling Lake Thun. This work has been supported by the Czech Science Foundation (CSF) through grants 19-23469S (awarded to M.M.S., supporting P.L., T.S. and M.M.S.), 20-12496X (awarded to R.G., supporting P.-A.B. and R.G.), 21-21990S (awarded to M.H., supporting M.-C.C. and M.H.), 22-03662S (awarded to M.M.S., supporting C.F., M.-C.C., and M.M.S.), 25-15813S (awarded to M.M.S., supporting C.F. and M.M.S.) and the Grant Agency of the University of South Bohemia in České Budějovice through grants 022/2019/P (supporting P.L.) and 017/2022/P (supporting C.F.).

## Author contributions

M.M.S. conceived and designed the study; M.M.S., P.L., M.H., R.G., T.S., V.L., C.C., B.S., T.P., F.L. and P.Z. sampled the lakes; M.M.S., P.L., C.F. and M.H. cultivated the strains; M.M.S., M.-C.C., P.-A.B., C.F. and M.H. performed bioinformatic and statistical analyses with software designed by P.-A.B. and R.G.; M.M.S. wrote the paper with contributions from all authors.

## Competing interests

Authors declare that they have no competing interests.
