## [Transparent Peer Review file · Nature Communications]

Bringing the uncultivated microbial majority of freshwater ecosystems into culture

Corresponding Author: Dr Michaela Salcher

Version 0:

Reviewer comments:

Reviewer #1

(Remarks to the Author)

The paper by Salcher and colleagues describes a large effort to leverage dilution-to-extinction with defined media to isolate abundant prokaryotes in the epilimnia and metalimnia of lakes across central Europe, resulting in 627 pure cultures. The paper also sequences paired metagenomes and compares a metagenome-assembled genome (MAG) set to the isolate collection and compares the isolate and MAG collection to a large number of metagenomes from lakes worldwide. The paper shows that the isolate collection represents many of the most abundant and prevalent genera in lakes and includes two previously undescribed families and nine undescribed genera. Those novel taxa are named under the SeqCode.

The paper tackles an important problem in microbiology at a large scale and generally delivers on the goal to isolate abundant and novel organisms. Some specific strengths are: large scale; paired isolate collection and metagenomes; analysis of growth rate and oligo/copiotrophy, high-quality of the isolate genomes; isolation of some interesting microorganisms including the first non-methanotrophic member of the Methylophilales and some other difficult groups like Armatimonadota. I do have some critical comments for the authors' consideration. None of this is major, but I hope the authors take the comments seriously if they have a chance to resubmit.

- In the title, please be aware that many scientists don't like the term "microbial dark matter". Also, I personally use the term only for higher ranks without cultured representatives (e.g., phyla, classes). But I'll leave the decision up to you.
- Line 18: evogenomic is not a standard word and is not a widely used term. Consider using evolutionary genomic. Also, please consider describing the importance of pure cultures more coherently. All scientist value pure samples (minerals, elements, spectra, etc.). It's easy to understand pure samples. Think of Koch's postulates for example.
- In the introduction, the discussion of auxotrophies and dependencies is a little off-target because this paper does not provide a solution to metabolic dependencies. The cells are diluted from the fresh samples into defined media with simple carbon sources and vitamins and other resources. Co-cultures are ignored, so this would prevent the study from growing microbes with metabolic dependencies (other than those supported by added media ingredients). Instead, I think the approach is very good at targeting slow-growing oligotrophs that can be outcompeted by copiotrophs during enrichment, which is not explicitly mentioned.
- Line 80, I suggest adding some text about novel biochemical pathways, cell ultrastructure, and other features here. The listed features are at least partly predictable from genomes (e.g., temperature optimum) and relatively boring. The paper would be improved by more creativity here.
- Line 108, what does "lineage-like taxa" mean?
- Line 124, how were the isolates divided into the three categories (oligotrophs, mesotrophs, copiotrophs)? Do the isolates form a continuum that was arbitrarily divided into thirds? Or were these discrete categories that were formed by a clustering method?
- Line 126, Fimbriococcus appears to be a new genus that is first described here. For all such new taxa, please clearly communicate this the first time the new names are mentioned.
- Line 131, there is a poor transition here from discussing oligotrophs to copiotrophs. A separate paragraph with topic sentence would help.
- Line 139, what is NSY?
- Figure 1 A and B, please define all elements of the box and whisker plots and give precise p-values (per Nature guidelines). Do this throughout the paper.

- Figure 1 B top, I don't understand what is plotted here. The labels refer to media but the legend refers to experiments with 18 different strains. So, was each strain only tested on one medium? There are only 18 data points (means presumably?) shown. Explain. Per Nature guidelines, show all data points and define all items on the graphs explicitly.
- Line 166 "significantly similar results" is awkward but more importantly what statistical analysis is this referring to? I think you would report just a percent of strains with the same phylum assignment? I was confused about the methods here.
- Line 170 and elsewhere (e.g., line 186, line 205), here you refer to genera as "taxa". Please try to be precise throughout the paper. In some cases, taxa is fine because a list transcends ranks, but please refer to genera or phyla or other specific ranks where possible.
- Line 175, here you refer to species not listed in LPSN as not "cultivated before" but it's really better to be precise. Most probably have been cultivated in some way but this number refers to species names that have not been validated under the ICSP (and presumably there are no pure cultures or stable co-cultures).
- Line 187 and elsewhere, the results are great but there's still a big cultivation bias toward Pseudomonadota, which is displayed nicely in Figure S7 B and C. I suggest you point this out clearly. For example, you could mention how many of these abundant genera axenic cultures are from Pseudomonadota and how many are from other phyla. I don't want to diminish the success of your work but just to communicate clearly that there's still a strong bias here and still a big problem with cultivation. Overall, as mentioned earlier, I think your study is very good for isolating oligotrophs that are outgrown in enrichment cultures or grow poorly on plates but it is poor at growing microbes with metabolic dependencies. It's ok, but it's good to be clear about the strengths of the study and where the study isn't so great (i.e., future directions).
- Figure 2C top, what is "Diversity" here? I can't understand what this is given the two y-axis labels, figure legend, and text.
- Figure 2C bottom, does this graph include read recruitment to MAGs as described in the text (I think)? If so, I think this is really off-target. What you need to show is the same graph with your isolate genomes or ideally a comparison of your isolate genomes and the complete genome collection (isolate genomes plus MAGs). Making just one plot with isolate genomes and MAGs does not communicate the significance (abundance) of the isolate collection, which is what this paper is about.
- Line 228 – beginning of the sentence needs fixing.
- Line 234 – please cite a research paper (or proper review) instead of a Microbe article. You could use reference 97, for example. Also, please clarify in the text here what ANI and AAI levels were used to indicate novel species and genera – presumably 95% and 65%?
- Line 240 - Use of the SeqCode is fine but please consider depositing the strain whose genome is the nomenclatural types into a culture collection because this could be an important resource for the research community. Please confirm in a response to reviewers that the entries in the SeqCode Registry have been pre-approved by curators.
- Figure 3 legend – For the full figure, make sure you refer to cultures and MAGs. For panel A, make sure it's clear in the figure legend that this is isolates only. For panel B, "gained" should be "assembled". In B, it's ok I guess but some parts of the figure are unreadable. There are too many categories for bootstrap values, genome size, and GC%. Categories should be lumped so there's a chance to see something. That could be done for coverage too. I know it's hard to deal with so much data. It's up to you if you want to address this, but just a suggestion.
- line 282 - 287 – more abundant than what? What does the text about "a significant distinction" mean? I can't understand it. Do you mean that larger genomes encoded [flagellar] motility and a larger number of different secretion systems? In the supplemental figures, it looks like you're referring to some genomes without ANY secretion system but as far as I know, all organisms have at least one secretion system, often either Sec or SecA2. If you're inferring that some genomes have no secretion system, that would be an extraordinary claim that would require extraordinary evidence.
- Line 288-312 is a repetition of the two preceding paragraphs.
- Line 314 – 315. First, not all rhodopsins are light-driven ion pumps that can make ion gradients (see channelrhodopsins and sensory rhodopsins). Please consider trying to distinguish these or qualify your statement. Second, I notice a lot of cyanobacteria isolates. Is this right? What about those?
- Line 316 – not "energy generation" but energy conservation – please. First Law of Thermodynamics.
- Line 318 – not all RuBisCO homologs participate in the CBB cycle. Please clarify the type.
- Line 321 – "organic carbon".
- Line 322 – what about catalase?
- Line 324 – Is this significant? Can you run a t-test for the endpoint?
- line 370 = ranks
- Line 406 – consider mentioning earlier in the main narrative that important tricks were used to increase cultivability, like catalase and pyruvate. I think these tricks aren't well known enough and it would be good to emphasize that you were careful and thoughtful and to help educate the community.
- Line 419 – all additions after autoclaving were filter sterilized? Please explain.
- Please check and confirm that the SeqCode Registry curators have provisionally accepted the proposed names and supporting data.

Additional comments on figures

Figure 1. the label "med3 (n=60)" was odd as shown in the figure; mention the statistical tests used.

Figure 2. The figure labels are overlapping. For Panel A, mention what the thicker and darker colors represent. For Panel C, add statistical test information.

Figure 3. The title "Diversity and phylogeny of genome-sequenced cultures" is not appropriate because the dataset also includes MAGs and other published genomes.

Supplementary figures

- Figure 1: Please show two-letter codes for lakes here and in Table S1.

- Figure 12 – define the parts of the boxes and whiskers. What is "Diversity"? Just to confirm – these are based on the culture genomes only and not the MAGs?

Very minor editorial suggestions

Introduction

Please check the citation formatting for the whole paper — there are extra spaces in some places.

Line 38: When "however" connects two independent clauses (complete sentences), use a semicolon before it and a comma after it.

Line 38: miniscule → minuscule

Line 46: nutrient rich → nutrient-rich

Line 48: nutrient rich → nutrient-rich

Line 69: the usage of → the use of

Line 86: missed a comma before "mimicking"

Line 89: Spring, Autumn, Summer → spring, autumn, summer

Results and Discussion

What are the differences between Supplementary Table 1 and Data S1?

Line 98: 96-deep-well-plates → 96-deep-well plates

Line 98: remove extra space after "one cell per well".

Line 102: maintained further → maintained thereafter

Line 109: contained → contain

Line 113: failed in obtaining → failed to obtain

Lines 117-121: the sentence is clunky; better to restructure.

Line 127: use a multiplication sign (×), not the letter "x".

Line 131: write out the full name of NSY when it is first mentioned.

Line 161: as used for → used for

Line 163: affiliated to → affiliated with

Line 213: of our culture collection → from our culture collection

Line 228: our the culture collection → from our culture collection

Line 232: the remaining ones → the remaining genomes

Line 225: a global presence → their global distribution

Line 234: Only six genomes could be assigned

Line 290: how did you select the 324 genomes of the closest relatives from public databases?

Line 293: Maybe you can highlight here that >83.9% are novel/undescribed species.

Line 376: differ to → differ from

Methods

Line 544: as an outgroup

Line 547: specify the number of bootstrap replicates

Reviewer #2

(Remarks to the Author)

Reviewer #3

(Remarks to the Author)

Salcher et al. present an innovative, high-throughput method to generate axenic cultures of aquatic prokaryotes. In doing so, they have addressed a major challenge in microbiology, i.e., "the great plate anomaly", that is the inability to cultivate a majority of microbial lineages. All the cultures presented here have been taxonomically annotated, and a subset had their genomes sequenced. The genome sequencing revealed diverse potential metabolisms, and as the cultures are now available, the genome-based predictions can be tested. Importantly, the cultured prokaryotes were among the most abundant in their environments, unlike many other, especially traditional cultivation techniques. Moreover, Salcher et al. used artificial media, which makes the cultures more transferable and easier to maintain as opposed to using autoclaved water.

Major comment:

As the authors correctly state in the Abstract, this study has resulted in a valuable collection of abundant freshwater microbes that holds significant potential as model systems for a wide array of studies. However, it is not stated how other researchers can get access to the isolates to perform such studies. This needs to be specified, otherwise this claim can hardly be made.

Minor comments / suggested improvements:

The detailed metabolic-capability map (Fig.4) for the 87 genome-sequenced strains could be combined with information on growth on the six media used in the study. The use of different media is a psychological experiment. Particularly, a medium

with methanol and methylamine (MM-med) was used, and the potential for methylotrophy was analyzed based on the genomes. Comparing the growth on the medium and the genome-inferred metabolic potential would be a major addition both to the analysis of the inferred metabolism and to showcasing the potential of the cultivation approach presented here.

The genomic comparison on page 10 and 11 is very informative and concise. An additional genomic feature interesting to compare would be CRISPRs. CRISPRs are known to be underrepresented in streamlined vs. non-streamlined genomes (for biological reasons) as well as in MAGs vs. isolates (for technical reasons), so comparing contents of CRISPR (eg proportion of genomes where CRISPRs are found) between your oligotrophic and copiotrophic isolates, as well as between your isolates and their closest MAG relatives (in cases where close MAG relatives exist) would be interesting. The latter comparison could also involve the number of CRISPR spacers found, since assembling MAGs may only recover the more conserved parts of the CRISPR within a population, as opposed to when assembling an isolate genome.

Putting supplementary figure S2 (and potentially also S1) in the main is worth consideration, since there is space for extra display items. Fig. S2 visualizes the workflow, which is the key new development presented in the paper. This is a fully suggestive comment, and it is ultimately up to the authors how they want to present their work.

Line 81: I find it very difficult to understand that phenotypic traits sometimes can be “not at all encoded in the genome”. Of course, if, for example, there is a phage infection, the phenotype may differ from a non-infected population. But also propensity for infection by a specific phage is encoded in the genome. So please specify how you mean this might work.
Line 89: It is unclear which year the spring and autumn samples come from (this is specified in the methods section and just requires a reformulation in the introduction for clarity).

Line 122: “Most cultures showed stable growth for more than one year” - the information on which/how many cultures showed stable growth for more than one year is not in the figures. If the details of long-term survival and growth are in a supplementary table, they should be referred to. Otherwise, at least provide the percentage or number of stable long-term cultures (or which cultures were unstable and/or hard to maintain).

Line 138: Please give the carbon source as concentration in the medium rather than as a mass.

Lines 147-156 (Fig. 3): The test used to obtain significance values is unspecified.

Line 166: It's odd to provide the lower range of the p-values (“>”), maybe the authors meant to write “<”?

Lines 177-178: It would be relevant here to write how many of the 30 genera (if any) overlap with the 48 previously cultivated genera mentioned on lines 174-175.

Line 189: Although I see what you mean, I guess the culture collection hasn't actually made a “contribution” to the diversity and abundance of taxa in the lakes. Consider rephrasing.

Figure 2. Panel B includes so many taxa with similar colors that it is impossible to identify many taxa in the barplots. Consider only coloring a subset of the taxa with highest abundances.

Line 218: I suggest you write “up to 117 x coverage per Gb mapped data” for clarity.

Line 228: “Eighty-seven strains our the culture collection” - this sentence is agrammatical and thus hard to read. Should it be “in our” rather than “our the” ?

Lines 236-237: What does “described species” mean here? Is it the species described according to the standards of ICNP, or, e.g., present in GTDB? Convention suggests the first option, but it would be beneficial to specify. The same applies to “undescribed genera” in line 237.

Lines 265-266: Is Data S11/S12 the same as Supplementary Table S11/S12? If yes, the cross-references should be consistent throughout the paper.

Lines 268-271: “In all but four cases” - do you mean “For all but four of the eighty-seven genomes”? And what does “closely related” MAG (95% ANI)?

Line 270: Shouldn't it be “Fig. 3B” here?

Line 280: Couldn't more TAA and less TGA/TAG simply reflect the general tendency of streamlined genomes to have lower GC-content?

Line 284: “significance distinction” could be reformulated to stress in what way these genomes were distinct (generally smaller/larger).

Lines 288-312: These two paragraphs are repeated. (I.e., these lines are almost exactly the same as lines 263-287; the only difference between the two versions is the figures and supplementary material).

Line 360: Should be “Fig. 4”

Line 395: Was hypolimnion defined as deeper than 5m?

Line 438: Since viability is a crucial statistic for the paper, it would be preferable to reiterate the formula in this study.

Reviewer #4

(Remarks to the Author)

Version 1:

Reviewer comments:

Reviewer #1

(Remarks to the Author)

Thanks for your thoughtful responses to my suggestions and congratulations on a very nice study. I will endorse acceptance for publication, but if possible consider changing "72 distinct genus or lineage-like taxa" to something like "72 distinct genera or unnamed SILVA lineages".

- Brian Hedlund

Reviewer #3

(Remarks to the Author)

The comments raised by us have all been well addressed in the revised version of the manuscript.

Reviewer #4

(Remarks to the Author)

REVIEWER COMMENTS

Reviewer #1 (Remarks to the Author):

The paper by Salcher and colleagues describes a large effort to leverage dilution-to-extinction with defined media to isolate abundant prokaryotes in the epilimnia and metalimnia of lakes across central Europe, resulting in 627 pure cultures. The paper also sequences paired metagenomes and compares a metagenome-assembled genome (MAG) set to the isolate collection and compares the isolate and MAG collection to a large number of metagenomes from lakes worldwide. The paper shows that the isolate collection represents many of the most abundant and prevalent genera in lakes and includes two previously undescribed families and nine undescribed genera. Those novel taxa are named under the SeqCode.

The paper tackles an important problem in microbiology at a large scale and generally delivers on the goal to isolate abundant and novel organisms. Some specific strengths are: large scale; paired isolate collection and metagenomes; analysis of growth rate and oligo/copiotrophy, high-quality of the isolate genomes; isolation of some interesting microorganisms including the first non-methanotrophic member of the Methylocidiphilales and some other difficult groups like Armatimonadota. I do have some critical comments for the authors' consideration. None of this is major, but I hope the authors take the comments seriously if they have a chance to resubmit.

RESPONSE: Thank you very much for your overall positive and encouraging review comments. Please find a detailed response to your individual comments below.

- In the title, please be aware that many scientists don't like the term "microbial dark matter". Also, I personally use the term only for higher ranks without cultured representatives (e.g., phyla, classes). But I'll leave the decision up to you.

RESPONSE: We thank the reviewer for this comment; we agree that the term "microbial dark matter" is mainly used for higher taxonomic ranks. We have changed the title of our manuscript to "Bringing the uncultivated microbial majority of freshwater ecosystems into culture".

- Line 18: evogenomic is not a standard word and is not a widely used term. Consider using evolutionary genomic. Also, please consider describing the importance of pure cultures more coherently. All scientist value pure samples (minerals, elements, spectra, etc.). It's easy to understand pure samples. Think of Koch's postulates for example.

RESPONSE: We changed “eco- and evogenomic investigations” to: “Axenic cultures are essential for studying microbial ecology, evolution, and genomics.” We further changed “Despite their importance,...” to: “Despite the importance of pure cultures,...”. (L 19-20)

- In the introduction, the discussion of auxotrophies and dependencies is a little off-target because this paper does not provide a solution to metabolic dependencies. The cells are diluted from the fresh samples into defined media with simple carbon sources and vitamins and other resources. Co-cultures are ignored, so this would prevent the study from growing microbes with metabolic dependencies (other than those supported by added media ingredients). Instead, I think the approach is very good at targeting slow-growing oligotrophs that can be outcompeted by copiotrophs during enrichment, which is not explicitly mentioned.

RESPONSE: We thank the reviewer for this suggestion. We have rephrased this part to (L 57-60): “Further, largely unknown growth requirements, adaptations to low nutrient and substrate concentrations, a tendency of being outcompeted by copiotrophs during enrichment, and a free-living lifestyle preventing growth on solid surfaces such as agar plates, make the isolation and cultivation of slowly growing aquatic oligotrophs challenging.”

- Line 80, I suggest adding some text about novel biochemical pathways, cell ultrastructure, and other features here. The listed features are at least partly predictable from genomes (e.g., temperature optimum) and relatively boring. The paper would be improved by more creativity here.

RESPONSE: We thank the reviewer for this suggestion. We added a sentence listing additional features that rely on cultures (L 80-85): “However, genomes alone are not sufficient to characterize the ecology of microbial taxa, as many phenotypic traits (e.g., cell size, temperature, pH, salinity, or substrate ranges and optima) are hard to identify or not at all encoded in the genome³⁹. Further, cultures are a prerequisite to discover and characterize biochemical pathways^{25,40,41}, cell ultrastructure^{42,43}, growth requirements^{17,29}, and microbial interactions^{15,44} and are the basis for genetic manipulations^{45,46}.”

- Line 108, what does "lineage-like taxa" mean?

RESPONSE: The SILVA database of 16S rRNA gene sequences assigns taxonomic ranks not always in a coherent way that follows the phylum-class-order-family-genus-species scheme. This is due to the many sequences obtained from uncultivated microbes that are included in this database. Hence, some important microbes are classified as clades or lineages, e.g., SAR11 clade I, II, or III (family or genus-level classifications within *Pelagibacterales*), hgcl (*Nanopelagicales*, i.e., order-level classification), GKS98 freshwater group (*Zwartia*, genus-level classification). We used a well-maintained and customized

SILVA-ARB database with updated taxonomic ranks wherever possible (e.g., we split hgcl into the genera *Planktophila*, *Nanopelagicus* and other, so far undescribed *Nanopelagicales*, see Supplementary Data 3 and 9. However, some lineages assigned by SILVA are hard to split (e.g., NS11-12 and OPS 17 seem to be orders), as 16S rRNA genes are often not discriminative enough to distinguish different genera and for many orders or families in SILVA, branches containing only environmental 16S rRNA sequences are simply grouped in uncultured lineages (e.g., Bacteria/Actinobacteriota/Acidimicrobiia/Microtrichales/uncultured or Bacteria/Proteobacteria/Alphaproteobacteria/Rhodobacterales_Rhodobacteraceae_1/uncultured).

- Line 124, how were the isolates divided into the three categories (oligotrophs, mesotrophs, copiotrophs)? Do the isolates form a continuum that was arbitrarily divided into thirds? Or were these discrete categories that were formed by a clustering method?
RESPONSE: The strains form a continuum that we arbitrarily divided into the three categories based on manual comparison of their growth rates. As suggested, we now did clustering of max. growth rates (based on Euclidean distance with Ward's method), however, the optimum number of clusters is 4, as one strain had extraordinarily fast growth (*Flavobacterium rarum* TH-1). For ease of comparison, we included this strain in the category "copiotrophs" as there is no such term as "ultracopiotroph" or "extracopiotroph". The clustering method also introduced slight changes in our manual division, as two strains that we initially thought to be copiotrophs are now in the group of mesotrophs. However, as mentioned above, the strains surely form a continuum with easily recognizable extremes (oligo- vs. copiotrophs). We still find significant differences between the three groups. We have included a new Figure (Suppl. Fig. 6, see also below) and modified the text accordingly to (L 131-133): "Although the strains form a continuum in growth characteristics, they could be grouped into oligo- and copiotrophs and a group of strains with in-between features (mesotrophs) after clustering based on growth rates (Supplementary Fig. 6)."

Supplementary Fig. 6: Hierarchical clustering of maximum growth rates of different strains. **a** Hierarchical clustering dendrogram based on maximum growth rates of different strains. Clustering was performed using Euclidean distance and Ward’s minimum variance linkage method (ward.D2). Red rectangles delineate the clusters identified based on the silhouette analysis. Strains are color-coded at the base of the dendrogram according to trophic classifications. **b** Bar plot showing the average silhouette width for clusters (k=2 to 5) with k=4 corresponding to the highest average width. Raw data can be found in Supplementary Data 5.

- Line 126, *Fimbriicoccus* appears to be a new genus that is first described here. For all such new taxa, please clearly communicate this the first time the new names are mentioned.

RESPONSE: We changed the sentence to (L 134-138): “Oligotrophs affiliated with *Planktophila* (*Actinomycetota*) and the newly proposed genera *Acidimicrobilacustris* gen. nov. (*Actinomycetota*) and *Fimbriicoccus* gen. nov. (*Armatimonadota*) showed slow growth... “. We further added fam. nov., gen. nov., and sp. nov. whenever we mention a newly proposed family, genus or species (e.g., L 145, 151, 207-210, 218, 229-233, 276, 286-312) and throughout the Supplementary text.

- Line 131, there is a poor transition here from discussing oligotrophs to copiotrophs. A separate paragraph with topic sentence would help.

RESPONSE: We started the description of copiotrophs in a new paragraph.

- Line 139, what is NSY?

RESPONSE: NSY is a medium containing nutrient broth, soytone, and yeast extract. This information is now provided in brackets together with a reference (Hahn et al. 2004).

- Figure 1 A and B, please define all elements of the box and whisker plots and give precise p-values (per Nature guidelines). Do this throughout the paper.

RESPONSE: We added a description of the box plots of this figure (now Fig. 2) as follows: "Boxes indicate the 25th and 75th quantiles, medians are displayed by central lines, whiskers indicate the 5th and 95th quantiles, and individual values are displayed by open circles." We also included the results of all statistical tests in a new supplementary table (Supplementary Data 4) and refer to this in the figure legend.

- Figure 1 B top, I don't understand what is plotted here. The labels refer to media but the legend refers to experiments with 18 different strains. So, was each strain only tested on one medium? There are only 18 data points (means presumably?) shown. Explain. Per Nature guidelines, show all data points and define all items on the graphs explicitly.

RESPONSE: In this figure (now Fig. 2), we plotted average growth rates and growth rate ranges of oligo-, meso- and copiotrophic strains grown in different media. Every strain was tested in the six media, individual plots (growth curves, growth rates, max. abundances) are included as Supplementary Figs. 4 & 5.

We changed the legend for clarification to: "**b** Maximum growth rates and abundances of 18 strains grown in six different media with low to high carbon and nutrient content. Strains were grouped in oligo-, meso- and copiotrophs according to growth characteristics. Averages and standard deviation of maximum growth rates for oligotrophs ($n = 6$ strains), mesotrophs ($n = 6$) and copiotrophs ($n = 6$; upper panel) as well as boxplots of their maximum growth yields (lower panel) are shown. Boxes indicate the 25th and 75th quantiles, medians are displayed by central lines, whiskers indicate the 5th and 95th quantiles, and individual samples are displayed by open circles. Significant differences between these groups (t-tests) are indicated by asterisks (***) ($p < 0.001$). Individual plots and growth curves of all strains are shown in Supplementary Figs. 4 and 5 and clustering of growth rates in Supplementary Fig. 6. Raw data are provided in Supplementary Data 3 and 5, and results of statistical tests in Supplementary Data 4."

- Line 166 "significantly similar results" is awkward but more importantly what statistical analysis is this referring to? I think you would report just a percent of strains with the same phylum assignment? I was confused about the methods here.

RESPONSE: We did a simple regression analysis between the results obtained by using 59 single copy marker genes (SingleM) and a single, commonly used marker gene (16S rRNA). We reformulated the sentence to make this clear (L163-166): "Relative abundances based on taxonomic classification using 59 single copy marker genes were significantly correlated

with results obtained by using 16S rRNA genes as taxonomic markers ($R^2 = 0.9725$, $p < 0.0001$, Supplementary Notes, Supplementary Figs. 8-10, Supplementary Data 4, 6, 8)."

- Line 170 and elsewhere (e.g., line 186, line 205), here you refer to genera as "taxa". Please try to be precise throughout the paper. In some cases, taxa is fine because a list transcends ranks, but please refer to genera or phyla or other specific ranks where possible.

RESPONSE: We changed "taxa" throughout the manuscript to genera, species or strains (e.g., 171, 173, 179).

- Line 175, here you refer to species not listed in LPSN as not "cultivated before" but it's really better to be precise. Most probably have been cultivated in some way but this number refers to species names that have not been validated under the ICSP (and presumably there are no pure cultures or stable co-cultures).

RESPONSE: We have rephrased this sentence accordingly to (L173-176): "Only 39 of the 157 abundant genera have been validly published under the International Code of Nomenclature of Prokaryotes (ICNP) according to the List of Prokaryotic names with Standing in Nomenclature (LPSN³) and another seven were proposed as *Candidatus* genera (Supplementary Data 7)."

- Line 187 and elsewhere, the results are great but there's still a big cultivation bias toward Pseudomonadota, which is displayed nicely in Figure S7 B and C. I suggest you point this out clearly. For example, you could mention how many of these abundant genera axenic cultures are from Pseudomonadota and how many are from other phyla. I don't want to diminish the success of your work but just to communicate clearly that there's still a strong bias here and still a big problem with cultivation. Overall, as mentioned earlier, I think your study is very good for isolating oligotrophs that are outgrown in enrichment cultures or grow poorly on plates but it is poor at growing microbes with metabolic dependencies. It's ok, but it's good to be clear about the strengths of the study and where the study isn't so great (i.e., future directions).

RESPONSE: We introduced the cultivation bias towards *Pseudomonadota* in L 118 as follows: "Our culture collection was mainly dominated by *Pseudomonadota* (50 out of 72 genera, many of them so far uncultivated), however, we failed to obtain isolates from several abundant bacterial (*Chloroflexota*, *Planctomycetota*) and archaeal (*Thermoproteota*) phyla that primarily inhabit the deep hypolimnion of freshwater lakes^{11,51}."

- Figure 2C top, what is "Diversity" here? I can't understand what this is given the two y-axis labels, figure legend, and text.

RESPONSE: This figure (now Fig. 3c) is based on taxonomic assignment of metagenomic reads via SingleM (genus level classification) of the same water samples (original samples,

n=67) or in metagenomes from all over the world (global lakes, n=462). The upper panel (“Diversity”) is the proportion of genera that were represented by at least one strain of our culture collection.

We modified the figure legend to make this more understandable: “c Proportion of genera from the culture collection relative to the total number of genera (diversity; upper panels) and their summed up relative abundances (abundance; lower panels) in the sampled lakes (original samples, $n = 67$; left panels) and in 462 publicly available metagenomes from seven continents (global lakes, $n = 462$; right panels). Metagenomic reads were taxonomy-assigned with SingleM, proportions of genera included in the culture collection are separately shown for different water layers (epi- and hypolimnion) and total. Boxes indicate the 25th and 75th quantiles, medians are displayed by central lines, whiskers indicate the 5th and 95th quantiles, individual samples are displayed by open circles. Significant differences between epi- and hypolimnetic samples (t-tests) are indicated by asterisks (***) ($p < 0.001$). Raw data are provided in Supplementary Data 7 and 10, results of statistical test in Supplementary Data 4.”

- Figure 2C bottom, does this graph include read recruitment to MAGs as described in the text (I think)? If so, I think this is really off-target. What you need to show is the same graph with your isolate genomes or ideally a comparison of your isolate genomes and the complete genome collection (isolate genomes plus MAGs). Making just one plot with isolate genomes and MAGs does not communicate the significance (abundance) of the isolate collection, which is what this paper is about.

RESPONSE: Please see also our response to the comment above. Fig. 3c is based on taxonomic assignment of metagenomic reads via SingleM (genus level classification). SingleM provides the whole bacterial community composition present in each metagenome (here used to give the genus level community composition). We then determined how many of the identified genera are present in our culture collection (upper panel) and how abundant the isolated genera are in these samples (lower panel).

- Line 228 – beginning of the sentence needs fixing.

RESPONSE: We changed the sentence to (L 217): “We selected 87 strains from our culture collection for whole-genome sequencing, including...”

- Line 234 – please cite a research paper (or proper review) instead of a Microbe article. You could use reference 97, for example. Also, please clarify in the text here what ANI and AAI levels were used to indicate novel species and genera – presumably 95% and 65%?

RESPONSE: Thank you for this suggestion. We removed the Microbe article and refer to Konstantinidis et al. 2017, ISMEJ instead (now reference 64). We also included the ANI and AAI cutoffs used in the text as follows (L 222-224): “GTDB classification⁵ and ANI and AAI comparisons (Supplementary Table S9-S10)⁵⁷ indicated that the genomes represent 27

genera and 47 distinct species by using AAI cutoffs of 65% and ANI cutoffs of 95%, respectively.”

- Line 240 - Use of the SeqCode is fine but please consider depositing the strain whose genome is the nomenclatural types into a culture collection because this could be an important resource for the research community. Please confirm in a response to reviewers that the entries in the SeqCode Registry have been pre-approved by curators.

RESPONSE: We plan to deposit some of the type strains in culture collections in the near future, however, this is not trivial as most strains are hard to maintain (no growth on agar plates, slow growth, low maximum cell yield [see Supplementary Figs. 4, 5] ruling out OD measurements). Meanwhile, we make strains available to the scientific community upon personal request by email. We amended the Data availability statement as follows: “Strains from the culture collection can be requested by email to Michaela M. Salcher (michaelasalcher@gmail.com).” We further confirm that the new taxa have been endorsed by the SeqCode curators, here are the confidential reviewer links:
[redacted]

- Figure 3 legend – For the full figure, make sure you refer to cultures and MAGs. For panel A, make sure it’s clear in the figure legend that this is isolates only. For panel B, “gained” should be “assembled”. In B, it’s ok I guess but some parts of the figure are unreadable. There are too many categories for bootstrap values, genome size, and GC%. Categories should be lumped so there’s a chance to see something. That could be done for coverage too. I know it’s hard to deal with so much data. It’s up to you if you want to address this, but just a suggestion.

RESPONSE: We thank the reviewer for these suggestions. We modified the figure legend (now Fig. 4) as follows: “**Figure 4. Diversity and phylogeny of genome-sequenced cultures compared to metagenome-assembled genomes (MAGs) assembled from the same samples.** **a** Number of genome-sequenced cultures with standing taxonomy and novel taxa on different phylogenetic levels (family, genus, species). **b** Phylogenomic tree of 120 single copy marker protein sequences of bacterial families containing cultures ($n = 87$), MAGs assembled from the same water samples ($n = 313$), and closely related published genomes ($n = 324$) from cultures (ref. culture), MAGs (ref. MAG), and SAGs (ref. SAG). Five genomes of *Patescibacteria* were used to root the tree. Clades are colored on a family level and genera containing cultures are marked in different colors and numbers (1: *Leadbetterella*; 2: *Flavobacterium*; 3: *Fimbriicoccus* gen. nov.; 4: *Acidimicrobilacustris* gen. nov.; 5: *Mycobacterium*; 6: *Rhodoluna*; 7: *Planktophila*; 8: *Verrucolacustris* gen. nov.; 9: *Fontibacterium*; 10: *Sphingobium*; 11: *Sphingorhabdus*; 12: *Rhabdaerophilum*; 13: *Aquidulcibacter*; 14: *Caulobacter*; 15: *Tabrizicola*; 16: *Allotabrizicola* gen. nov.; 17: *Novimethylotenera* gen. nov.; 18: *Methylotenera*; 19: *Methylopumilus*; 20: *Lacustribacter* gen. nov.; 21: *Hahnella* gen. nov.; 22: *Zwartia*; 23: *Hydrogenophaga*; 24: *Allorhodoferax*”).

gen. nov.; 25: *Limnohabitans*; 26: *Pernthalerella* gen. nov.; 27: *Polynucleobacter*). Annotations of the rings from inside to outside: Genome source; genome size; genomic GC content; metagenomic fragment recruitment for 67 samples where cultures and MAGs were gained. The lakes are sorted from oligo- to eutrophic as in Fig. 3 and epi- and hypolimnetic samples are shown separately. See Supplementary Fig. 14 for a full tree with all 1,294 dereplicated MAGs covering all families.”

- line 282 - 287 – more abundant than what? What does the text about “a significant distinction” mean? I can’t understand it. Do you mean that larger genomes encoded [flagellar] motility and a larger number of different secretion systems? In the supplemental figures, it looks like you’re referring to some genomes without ANY secretion system but as far as I know, all organisms have at least one secretion system, often either Sec or SecA2. If you’re inferring that some genomes have no secretion system, that would be an extraordinary claim that would require extraordinary evidence.

RESPONSE: We are sorry for this confusion and changed the text to make it clear that genome-streamlined microbes were more abundant than microbes with medium to large genomes. We also clarified that microbes with genes encoding motility (both flagellar motility and gliding motility) and secretion systems type II, IV, or VI were larger in genome size than those lacking these traits. All genomes contained genes encoding Sec and Tat secretion systems; we didn’t specifically highlight this in the text, but this information can be found in Table S15. We changed the text to (L 256-262): “Genome-streamlined bacteria were also clearly more abundant than microbes with larger genomes, both in the original water samples ($n = 67$) and in freshwater metagenomes gained from six continents ($n = 250$). Further, microbes containing genes encoding motility and secretion systems type II, IV, or VI had significantly larger genome sizes (Supplementary Fig. 18, Supplementary Data 4, 18), except for two *Zwartia* strains with large genomes (3.6 and 4.5 Mbp) that lacked genes for flagella assembly and chemotaxis, congruent with previously described *Zwartia* strains⁶⁷.” We further added a sentence to the figure legend of Fig. 5 as follows: “Please note that general secretion systems Sec and Tat are not shown here as these were present in all strains.”

- Line 288-312 is a repetition of the two preceding paragraphs.

RESPONSE: Thank you for identifying this mistake, it happened in the last round of revision before submission when we decided to shuffle some parts of the text. The duplicated paragraph has been deleted.

- Line 314 – 315. First, not all rhodopsins are light-driven ion pumps that can make ion gradients (see channelrhodopsins and sensory rhodopsins). Please consider trying to distinguish these or qualify your statement. Second, I notice a lot of cyanobacteria isolates. Is this right? What about those?

RESPONSE: We only searched for proton-pumping rhodopsins in this study and most taxa contained either actino-, proteo- or bacteriorhodopsins. We changed the text accordingly to (268-270): “Most genome sequenced cultures can potentially utilize light as an energy source, either *via* proton-pumping rhodopsins ($n = 59$) or by being anoxygenic aerobic phototrophs (AAPs, $n = 15$).” Regarding the second question, while we obtained many cyanobacterial strains, we did not genome-sequence them as this study focuses on heterotrophs.

- Line 316 – not “energy generation” but energy conservation – please. First Law of Thermodynamics.

RESPONSE: We are sorry for this rookie mistake; we changed the text accordingly.

- Line 318 – not all RuBisCO homologs participate in the CBB cycle. Please clarify the type.

RESPONSE: The five AAPs mentioned in the text contain RuBisCO type I, which is participating in the CBB cycle. Additionally, one strain (*Tabrizicola rara* LH-M10) contained RuBisCO type II, which is also participating in the CBB cycle and three strains contained RuBisCO-like proteins (RLP) type IV, which are not participating in CBB. We show only RuBisCO types I and II in Fig. 5. We added this information also in the text (L 273-275): “Interestingly, five AAPs additionally contained a carbon fixation pathway (Calvin cycle via RuBisCO type I), potentially leading to photoautotrophy that was so far only occasionally reported for sediment dwelling AAPs⁷²”

- Line 321 – “organic carbon”.

RESPONSE: Changed.

- Line 322 – what about catalase?

RESPONSE: We did not add catalase to the medium used for this experiment, only vitamins.

- Line 324 – Is this significant? Can you run a t-test for the endpoint?

RESPONSE: We ran a paired t-test for the endpoint; it was significantly different ($p = 0.0092$, test results are given in Supplementary Data 4). We added this information in the text, it now reads (L 278-281): “While growth was observed in both dark and light:dark (12:12 h) conditions, *L. simekii* maintained significantly higher cell numbers during stationary phase in light:dark conditions ($p = 0.009175$, Supplementary Data 4) hinting at photoautotrophic growth during starvation.”

- line 370 = ranks

RESPONSE: Changed.

- Line 406 – consider mentioning earlier in the main narrative that important tricks were used to increase cultivability, like catalase and pyruvate. I think these tricks aren't well

known enough and it would be good to emphasize that you were careful and thoughtful and to help educate the community.

RESPONSE: We added this information in the last paragraph of the introduction (L 88-92): “Here, we employed a high-throughput dilution-to-extinction approach with three defined artificial media containing either different carbohydrates, organic acids, catalase, vitamins, and other organic compounds in μM concentrations, mimicking carbon concentrations typically found in freshwater lakes (med2 and med3), or methanol, methylamine and vitamins (MM-med) as sole carbon sources (Supplementary Data 1).” We further included media components and preparation as Supplementary Data 1.

- Line 419 – all additions after autoclaving were filter sterilized? Please explain.

RESPONSE: Yes, we filtered all components through 0.1 μm filters, this information is now also mentioned in the methods as follows (L 368-370): “After autoclaving, both media were amended with a filter-sterilized mix of the 20 proteinogenic amino acids (0.2 μM of each amino acid except for glutamate and glutamine which were added at 0.4 μM concentration), vitamins (0.593 μM thiamine, 0.08 μM niacin, 0.074 nM cobalamine, 0.005 μM para-amino benzoic acid, 0.074 μM pyridoxine, 0.081 μM pantothenic acid, 0.004 μM biotin, 0.004 μM folic acid, 0.555 μM myo-inositol), dipotassium phosphate (3.22 μM) and catalase (10 U ml^{-1})⁸⁶.” We further included media components and preparation as Supplementary Data 1.

- Please check and confirm that the SeqCode Registry curators have provisionally accepted the proposed names and supporting data.

RESPONSE: We confirm that the new taxa have been endorsed by the SeqCode curators (see above for confidential reviewer links)

Additional comments on figures

Figure 1. the label “med3 (n=60)” was odd as shown in the figure; mention the statistical tests used.

RESPONSE: We updated the figure (now Fig. 2) as requested and mention the statistical tests (t-tests) in the figure legend. We further refer to Supplementary Data 4 for details on all statistical tests.

Figure 2. The figure labels are overlapping. For Panel A, mention what the thicker and darker colors represent. For Panel C, add statistical test information.

RESPONSE: We have reorganized this figure (now Fig. 3) to make it more appealing. For panel A, we specified that the thick lines are averages and the thin lines are maxima as follows: “a Rank-abundance curve (lower panel, averages are shown as thick lines, maxima as thin lines) and number of axenic cultures (upper panel) of abundant genera present in the sampled lakes (>1% of reads in at least one sample).” For panel C, we now included

statistical test information as follows: **c** Proportion of genera from the culture collection relative to the total number of genera (diversity; upper panels) and their summed up relative abundances (abundance; lower panels) in the sampled lakes (original samples, $n = 67$; left panels) and in 462 publicly available metagenomes from seven continents (global lakes, $n = 462$; right panels). Metagenomic reads were taxonomy-assigned with SingleM, proportions of genera included in the culture collection are separately shown for different water layers (epi- and hypolimnion) and total. Boxes indicate the 25th and 75th quantiles, medians are displayed by central lines, whiskers indicate the 5th and 95th quantiles, individual samples are displayed by open circles. Significant differences between epi- and hypolimnetic samples (t-tests) are indicated by asterisks (***) ($p < 0.001$). Raw data are provided in Supplementary Data 7 and 10, results of statistical tests in Supplementary Data 4.”

Figure 3. The title “Diversity and phylogeny of genome-sequenced cultures” is not appropriate because the dataset also includes MAGs and other published genomes.

RESPONSE: We changed the title of this figure (now Fig. 4) as follows: “Figure 4. Diversity and phylogeny of genome-sequenced cultures compared to metagenome-assembled genomes (MAGs) assembled from the same samples”

Supplementary figures

- Figure 1: Please show two-letter codes for lakes here and in Table S1.

RESPONSE: We added the two-letter codes also here and in the supplementary table (now Supplementary Data 2)

- Figure 12 – define the parts of the boxes and whiskers. What is “Diversity”? Just to confirm – these are based on the culture genomes only and not the MAGs?

RESPONSE: This figure is based on taxonomic assignment of metagenomic reads via SingleM (genus level classification) of water samples from all over the world ($n=462$). The upper panel (“Diversity”) is the proportion of genera that were represented by at least one strain of our culture collection. We modified the figure legend to make this more understandable: “Supplementary Figure 12. Contribution of cultures to the total number (diversity; upper panel) and summed up read counts (abundance, lower panel) of genera present in 462 freshwater metagenomic samples from seven different continents.”

Very minor editorial suggestions

Introduction

Please check the citation formatting for the whole paper — there are extra spaces in some places.

RESPONSE: We carefully checked all citations throughout the manuscript and supplementary text and removed all extra spaces.

Line 38: When "however" connects two independent clauses (complete sentences), use a semicolon before it and a comma after it.

RESPONSE: Changed.

Line 38: miniscule → minuscule

RESPONSE: Changed.

Line 46: nutrient rich → nutrient-rich

RESPONSE: Changed.

Line 48: nutrient rich → nutrient-rich

RESPONSE: Changed.

Line 69: the usage of → the use of

RESPONSE: Changed.

Line 86: missed a comma before “mimicking”

RESPONSE: Changed.

Line 89: Spring, Autumn, Summer → spring, autumn, summer

RESPONSE: Changed.

Results and Discussion

What are the differences between Supplementary Table 1 and Data S1?

RESPONSE: Supplementary Table S1 and Data S1 are the same, we apologize for this misnomer. We have now renamed all tables as Supplementary Data 1, 2, 3, etc. as per the formatting instructions of Nature Communications.

Line 98: 96-deep-well-plates → 96-deep-well plates

RESPONSE: Changed.

Line 98: remove extra space after “one cell per well”.

RESPONSE: Changed.

Line 102: maintained further → maintained thereafter

RESPONSE: Changed.

Line 109: contained → contain

RESPONSE: Changed.

Line 113: failed in obtaining → failed to obtain

RESPONSE: Changed.

Lines 117-121: the sentence is clunky; better to restructure.

RESPONSE: We have split this sentence in two (L124-128): “The enrichment in a medium

with methanol and methylamine as sole carbon sources resulted in an enhanced cultivation of methylotrophs (*Methylophilus*, *Methylotenera*^{50,52}). However, also other genera were able to grow in this C1 medium (e.g., *Polynucleobacter*, *Limnohabitans*) and might have survived by either being photoautotrophic, methylotrophic, or by using catalase or vitamins as additional carbon sources.“

Line 127: use a multiplication sign (×), not the letter "x".

RESPONSE: Changed, also in lines 137, 140, 206, 210, 276, 357, 400, 402, 406-407, 415, and throughout the Supplementary Text

Line 131: write out the full name of NSY when it is first mentioned.

RESPONSE: We wrote that NSY is a medium containing nutrient broth, soytone, and yeast extract in brackets.

Line 161: as used for → used for

RESPONSE: Changed.

Line 163: affiliated to → affiliated with

RESPONSE: Changed.

Line 213: of our culture collection → from our culture collection

RESPONSE: Changed.

Line 228: our the culture collection → from our culture collection

RESPONSE: Changed.

Line 232: the remaining ones → the remaining genomes

RESPONSE: Changed.

Line 225: a global presence → their global distribution

RESPONSE: Changed.

Line 234: Only six genomes could be assigned

RESPONSE: Changed.

Line 290: how did you select the 324 genomes of the closest relatives from public databases?

RESPONSE: We first looked for cultivated type strains in GTDB (gtdb.ecogenomics.org) by verbatim name matching of genus names assigned to genome-sequenced cultures (Advanced Search, match all of the following: “GTDB Type Material” IS “type strain of species” AND “GTDB Taxonomy” CONTAINS “genus taxonomy of genome-sequenced strain”, e.g., *g_Zwartia*). We then added additional, not cultivated representative species in GTDB by verbatim name matching (Advanced Search, match all of the following: “GTDB

Representative of Species” IS TRUE AND “GTDB Taxonomy” CONTAINS “genus taxonomy of genome-sequenced strain”). In the case of novel families that we described in this manuscript, we extended this search to additional, closely related families within the same order (e.g., Acidimicrobiales). This list was amended with additional genomes identified via literature research based on genus or family names. We added this information in the material and methods part as follows (L 492-495): “Taxonomic classification was done with GTDB r220 as outlined above and the closest relatives of each culture genome ($n = 324$) were selected in GTDB by verbatim name matching of the assigned genus or family name and literature research and downloaded from NCBI.”

Line 293: Maybe you can highlight here that >83.9% are novel/undescribed species.

RESPONSE: We modified the sentence accordingly (L238-240): Only seven MAGs could be assigned to a described species and another seven to *Candidatus* species (1.1% of the 1,294 MAGs), reiterating the high proportion of undescribed taxa in freshwater environments^{4,11}.

Line 376: differ to → differ from

RESPONSE: Changed.

Methods

Line 544: as an outgroup

RESPONSE: Changed.

Line 547: specify the number of bootstrap replicates

RESPONSE: We did 1000 ultrafast bootstrap replicates; this information is now included in the text (L 511).

Reviewer #2 (Remarks to the Author):

RESPONSE: Thank you very much for your review. We highly appreciate the Nature Communications initiative to facilitate training in peer review for Early Career Researchers.

Reviewer #3 (Remarks to the Author):

Salcher et al. present an innovative, high-throughput method to generate axenic cultures of aquatic prokaryotes. In doing so, they have addressed a major challenge in microbiology, i.e., “the great plate anomaly”, that is the inability to cultivate a majority of microbial lineages. All the cultures presented here have been taxonomically annotated, and a subset had their genomes sequenced. The genome sequencing revealed diverse potential metabolisms, and as the cultures are now available, the genome-based predictions can be tested. Importantly, the cultured prokaryotes were among the most abundant in their environments, unlike many other, especially traditional cultivation techniques. Moreover, Salcher et al. used artificial media, which makes the cultures more transferable and easier to maintain as opposed to using autoclaved water.

RESPONSE: Thank you very much for your overall positive and encouraging review comments. Please find a detailed response to your individual comments below.

Major comment:

As the authors correctly state in the Abstract, this study has resulted in a valuable collection of abundant freshwater microbes that holds significant potential as model systems for a wide array of studies. However, it is not stated how other researchers can get access to the isolates to perform such studies. This needs to be specified, otherwise this claim can hardly be made.

RESPONSE: We thank the reviewer for this comment and apologize that we forgot to mention this information in the initial submission. All strains from our culture collection were cryopreserved and are available to other researchers by email request to the corresponding author (michaelasalcher@gmail.com). We have added this information to the Data availability statement as follows (L 532-533): “Strains from the culture collection can be requested by email to Michaela M. Salcher (michaelasalcher@gmail.com).”

Minor comments / suggested improvements:

The detailed metabolic-capability map (Fig.4) for the 87 genome-sequenced strains could be combined with information on growth on the six media used in the study. The use of different media is a psychological experiment. Particularly, a medium with methanol and methylamine (MM-med) was used, and the potential for methylotrophy was analyzed based on the genomes. Comparing the growth on the medium and the genome-inferred

metabolic potential would be a major addition both to the analysis of the inferred metabolism and to showcasing the potential of the cultivation approach presented here.

RESPONSE: We did not use the MM-med in growth assays with known methylotrophs such as *Methylopumilus*, as this was already done in earlier studies (Salcher et al. 2015 ISMEJ, Layoun et al. 2024, ISMEJ). As for the other media used in the growth assays, it is not straightforward to match the carbon sources to growth patterns. For example, med2 and med3 are very similar in C composition, med3 additionally contains glycolate, polyamines and two carbohydrates compared to med2. We checked for the presence of glycolate oxidation in the genomes, but there was no obvious pattern connected to growth in med3, i.e., six strains grew better in med3 than in med2, but only two of them (*Polynucleobacter hoetzingerianus* RE-M21 and *Allorhodofera lacus* MsE-M18) contained pathways for glycolate oxidation, while the others (*Acidimicrobiolacustris europeus* KE-4, *Planktophila versatilis* MsE-18, *Fimbriicoccus planktonicus* MsE-15, *Flavobacterium raram* TH-M1) did not. The same holds true for polyamines, only four of the strains that grew better in med3 contained membrane transporters for spermidine or putrescine (*A. europeus* KE-4, *P. versatilis* MsE-18, *A. lacus* MsE-M18, *F. raram* TH-M1). We tried different versions of how to integrate the different media in this figure (Fig. 5), but none of them looked compelling, therefore we decided to keep it as is.

The genomic comparison on page 10 and 11 is very informative and concise. An additional genomic feature interesting to compare would be CRISPRs. CRISPRs are known to be underrepresented in streamlined vs. non-streamlined genomes (for biological reasons) as well as in MAGs vs. isolates (for technical reasons), so comparing contents of CRISPR (eg proportion of genomes where CRISPRs are found) between your oligotrophic and copiotrophic isolates, as well as between your isolates and their closest MAG relatives (in cases where close MAG relatives exist) would be interesting. The latter comparison could also involve the number of CRISPR spacers found, since assembling MAGs may only recover the more conserved parts of the CRISPR within a population, as opposed to when assembling an isolate genome.

RESPONSE: We thank the reviewer for this great suggestion. We predicted CRISPR-Cas systems in our culture collection, MAGs and reference genomes and found that only 5 strains (5% of the culture collection) with medium-large genome sizes contained this phage-defense system. In contrast, ~20% of the reference genomes obtained from cultures and ~13% of our MAGs contained CRISPR-Cas repeats. Further, as could be expected, there was a clear relationship between genome size and CRISPR system (with larger genomes being more likely to have CRISPR). We have made an additional Supplementary Figure (Suppl. Fig. 19, see below) and table (Supplementary Data 19) and included the following text in the manuscript (L 262-267): “The phage-defense system CRISPR-Cas^{68,69} was present in only 5 strains with medium-large genome sizes (2.6-5 Mbp, Supplementary Fig. 19, Supplementary Data 19). In contrast, 20.8% of the reference genomes obtained from cultures and 13.9% of MAGs contained on average 1.7 CRISPR

arrays per genome. There was a clear relationship to genome size, as microbes containing CRISPR-Cas systems were significantly larger than those without ($p = 7.524E-47$, Supplementary Data 4).”

Supplementary Figure 19. Occurrence of CRISPR-Cas arrays in cultures, MAGs and closely related cultivates references. **a** The same phylogenomic tree as in Fig. 4 including the number of CRISPR-Cas arrays per genome. **b** Number of genomes with CRISPR-Cas arrays and number of CRISPR-Cas arrays per genome separately shown for cultures,

reference cultures, and MAGs or **c** different genome size classes. **d** Genome sizes of microbes with or without CRISPR-Cas arrays. *** $p < 0.0001$ for significant differences in genome sizes (t-test), results of statistical tests can be found in Supplementary Data 4.

Putting supplementary figure S2 (and potentially also S1) in the main is worth consideration, since there is space for extra display items. Fig. S2 visualizes the workflow, which is the key new development presented in the paper. This is a fully suggestive comment, and it is ultimately up to the authors how they want to present their work.

RESPONSE: We thank the reviewer for this valuable suggestion. We have now included Fig. S2 as a new main figure (new Fig. 1).

Line 81: I find it very difficult to understand that phenotypic traits sometimes can be “not at all encoded in the genome”. Of course, if, for example, there is a phage infection, the phenotype may differ from a non-infected population. But also propensity for infection by a specific phage is encoded in the genome. So please specify how you mean this might work.

RESPONSE: Here, we argue that some traits can be very hard to infer from genomes without having cultures (e.g., cell size, temperature, pH, salinity, or substrate ranges and optima). We expanded this part as follows (L780-85): “However, genomes alone are not sufficient to characterize the ecology of microbial taxa, as many phenotypic traits (e.g., cell size, temperature, pH, salinity, or substrate ranges and optima) are hard to identify or not at all encoded in the genome³⁹. Further, cultures are a prerequisite to discover and characterize biochemical pathways^{25,40,41}, cell ultrastructure^{42,43}, growth requirements^{17,29}, and microbial interactions^{15,44} and are the basis for genetic manipulations^{45,46}.”

Line 89: It is unclear which year the spring and autumn samples come from (this is specified in the methods section and just requires a reformulation in the introduction for clarity).

RESPONSE: The spring and autumn samples were also obtained in 2019. We have now included this information in the text (L 92-94) as follows: “We sampled 14 lakes in Central Europe during spring and autumn 2019, and four lakes additionally in summer 2019 (Supplementary Fig. 1, Supplementary Data 2).

Line 122: “Most cultures showed stable growth for more than one year” - the information on which/how many cultures showed stable growth for more than one year is not in the figures. If the details of long-term survival and growth are in a supplementary table, they should be referred to. Otherwise, at least provide the percentage or number of stable long-term cultures (or which cultures were unstable and/or hard to maintain).

RESPONSE: We have included the information about unstable cultures in the beginning of the results section (L103-106: “Screening resulted in 1,201 initial cultures, whereof 229 were identified as mixed by Sanger sequencing of 16S rRNA gene amplicons and 344 cultures showed no growth after several transfers and were discarded (Supplementary Fig.

2, Supplementary Data 3).”), in Supplementary Fig. 2 and in a Supplementary table (Supplementary Data 3, row labeled as “# cultures with no growth after 3-4 propagations”). However, it is very hard to reconstruct the exact time when we stopped propagating individual cultures, as we discontinued many strains from taxa for which we isolated multiple representatives after taxonomy assignment by 16S rRNA sequencing. This process took quite a long time (~1 year) because of severe restrictions in lab work during COVID19 lockdowns, which was also a reason why we kept only a reduced number of strains in this period. Selected strains were kept until we obtained their genomes (>1 year). However, multiple strains are being maintained till today (e.g., the *Fontibacterium* and *Planktophilia* strains) while some others have been revived several times from glycerol-stocks (e.g. for the growth assays).

Line 138: Please give the carbon source as concentration in the medium rather than as a mass.

RESPONSE: We have reformulated this sentence to (L 147-150): “Copiotrophic strains grew to highest densities in the medium with highest nutrient content (1:10 diluted NSY containing 0.3 g complex carbon sources per liter), and several of these genera have been previously isolated by the filtration acclimatization method using NSY⁵³.” We further added an additional table (Supplementary Data 1) with details on all media used for isolation and for growth assays.

Lines 147-156 (Fig. 3): The test used to obtain significance values is unspecified.

RESPONSE: We used t-tests to test for significant differences; this is now included in the figure legend. We further added all results of statistical test to a Supplementary Table (Supplementary Data 4).

Line 166: It’s odd to provide the lower range of the p-values (“>”), maybe the authors meant to write “<”?

RESPONSE: We thank the reviewers for finding this typo, it is changed to “<” now.

Lines 177-178: It would be relevant here to write how many of the 30 genera (if any) overlap with the 48 previously cultivated genera mentioned on lines 174-175.

RESPONSE: We have added this information to the text and also in Supplementary Data 7 as follows (L179-180): “Only 16 of the 30 abundant genera of our culture collection have been validly described to date³.”

Line 189: Although I see what you mean, I guess the culture collection hasn’t actually made a “contribution” to the diversity and abundance of taxa in the lakes. Consider rephrasing.

RESPONSE: We rephrased the legend accordingly to: **“Figure 3. Representation of the culture collection in lake samples. a Rank-abundance curve (lower panel, averages are**

shown as thick lines, maxima as thin lines) and number of axenic cultures (upper panel) of abundant genera present in the sampled lakes (>1% of reads in at least one sample). Metagenomic reads were taxonomy-assigned with SingleM. Asterisks below cultures indicate that at least one member of the genus was genome-sequenced. See Supplementary Fig. 11 and Supplementary Data 7 for all genera including rare taxa. **b** Summed up relative abundances of taxa with representatives in the culture collection in the epi- and hypolimnion of the sampled lakes. Samples are sorted from oligo- to eutrophic, abbreviations of lakes and sampling seasons are as in Supplementary Fig. 1. **c** Proportion of genera from the culture collection relative to the total number of genera (diversity; upper panels) and their summed up relative abundances (abundance; lower panels) in the sampled lakes (original samples, $n = 67$; left panels) and in 462 publicly available metagenomes from seven continents (global lakes, $n = 462$; right panels). Metagenomic reads were taxonomy-assigned with SingleM, proportions of genera included in the culture collection are separately shown for different water layers (epi- and hypolimnion) and total. Boxes indicate the 25th and 75th quantiles, medians are displayed by central lines, whiskers indicate the 5th and 95th quantiles, individual samples are displayed by open circles. Significant differences between epi- and hypolimnetic samples (t-tests) are indicated by asterisks (***: $p < 0.001$). Raw data can be found as Supplementary Data 7 and 10, results of statistical test as Supplementary Data 4.”

Figure 2. Panel B includes so many taxa with similar colors that it is impossible to identify many taxa in the barplots. Consider only coloring a subset of the taxa with highest abundances.

RESPONSE: We agree that we used too many different colors in this figure. We summed up several taxa (e.g., *Aquiluna* & *Rhodoluna*, UBA2463 & UBA954, *Limnohabitans* & *Limnohabitans_A*,...) to make the figure less crowded. Please note that we have modified this figure (now Fig. 3) also based on suggestions by reviewer #1.

Line 218: I suggest you write “up to 117 x coverage per Gb mapped data” for clarity.

RESPONSE: Changed as suggested.

Line 228: “Eighty-seven strains our the culture collection” - this sentence is agrammatical and thus hard to read. Should it be “in our” rather than “our the” ?

RESPONSE: Thank you for finding this mistake. We changed it to “We selected 87 strains from our culture collection for whole-genome sequencing” (L 217)

Lines 236-237: What does “described species” mean here? Is it the species described according to the standards of ICNP, or, e.g., present in GTDB? Convention suggests the first option, but it would be beneficial to specify. The same applies to “undescribed genera” in line 237.

RESPONSE: We have modified the text to make the distinction between validly described and novel taxa. The text reads now as follows (L224-228): “Only one genome could be

assigned to a validly described species (*Sphingobium curpirestens*⁶⁵) and five more were previously proposed as *Candidatus* species by us^{24,52} (Supplementary Fig. 17). Moreover, genome-sequenced cultures contained nine novel genera and two novel families (Fig. 4a; Supplementary Data 11).”

Lines 265-266: Is Data S11/S12 the same as Supplementary Table S11/S12? If yes, the cross-references should be consistent throughout the paper.

RESPONSE: Supplementary Table S11/S12 and Data S11/S12 are the same, we apologize for this misnomer. We have now unified the naming of all Supplementary tables to Supplementary Data 1-21 as per the formatting instructions of Nature Communications.

Lines 268-271: “In all but four cases” - do you mean “For all but four of the eighty-seven genomes”? And what does “closely related” MAG (95% ANI?)

RESPONSE: We calculated ANI values between our genome-sequenced strains and both the closest MAG and publicly available genome-sequenced culture. We have reanalyzed the data and have now included an additional figure that illustrates that the vast majority of our strains is closer related to environmental MAGs than to cultures (Suppl. Fig. 17, see also below). Moreover, by taking 95% ANI as species cutoff, we show that only one of our strains has a cultivated relative that is validly described and available in a strain collection (*Sphingobium cuppiresistens* CCTCC AB 2011146), while another culture genome (*Limnohabitans* sp. B9-3) was not validly described and five more *Candidatus* species were previously proposed by us (3 *Methylopumilus* species: Salcher et al. 2015 ISMEJ, Salcher et al. 2019 ISMEJ, 2 *Planktophila* species: Neuenschwander et al. 2018 ISMEJ). We rephrased this sentence to (L240-244): “In the majority of cases, strains from our culture collection were phylogenetically closer to MAGs than to previously cultured species, and only one, *Sphingobium cuppiresistens*, is validly described and available in a culture collection⁶⁵ (Fig. 4b, Supplementary Figs. 14, 17), exemplifying that the genome-sequenced cultures were indeed novel and highly relevant.”

Strain code:

- | | | |
|--|---|--|
| 1 Acidimicrobilacustris thunensis TE-4 | 30 Sphingorhabdus communis GE-11 | 59 Methylopusillus planktonicus MoH-M8 |
| 2 Acidimicrobilacustris thunensis TE-7 | 31 Sphingorhabdus communis ZE-10 | 60 Methylopusillus planktonicus MsH-M18 |
| 3 Acidimicrobilacustris europaeus KE-4 | 32 Hydrogenophaga miladensis MiE-M28 | 61 Methylopusillus planktonicus MsH-M39 |
| 4 Rhodoluna miladensis MiE-23b | 33 Limnohabitans simekii MiE-M12 | 62 Methylopusillus planktonicus MsH-M42 |
| 5 Rhodoluna miladensis MiE-24b | 34 Limnohabitans rimovensis RE-1 | 63 Methylopusillus planktonicus TH-M4 |
| 6 Mycobacterium aquicola MaE-M6b | 35 Limnohabitans kasalickyi MaE-M4 | 64 Methylopusillus planktonicus ZE-M7 |
| 7 Mycobacterium aquicola MiE-22 | 36 Polynucleobacter hoetzingianus RE-M21 | 65 Methylopusillus planktonicus ZE-M8 |
| 8 Planktophila dulcis MaH-2 | 37 Pernthalerella aquatica GE-M3 | 66 Methylopusillus rimovensis RE-M17 |
| 9 Planktophila dulcis MsH-2 | 38 Pernthalerella lacunae MaE-M21 | 67 Methylopusillus rimovensis RE-M20 |
| 10 Planktophila dulcis TrE-23 | 39 Pernthalerella communis MsE-6 | 68 Methylopusillus rimovensis RE-M24 |
| 11 Planktophila grossartii RE-3 | 40 Polynucleobacter hahnii ZE-4 | 69 Methylopusillus universalis GE-M14 |
| 12 Planktophila grossartii RH-3 | 41 Allorhodofera aquaticus MiE-M13 | 70 Methylopusillus universalis GH-M24 |
| 13 Planktophila turcensis ZE-9 | 42 Allorhodofera lacus MsE-M18 | 71 Methylopusillus universalis GH-M4 |
| 14 Planktophila warneckei RE-8 | 43 Allorhodofera aquaticus MsE-M22 | 72 Methylopusillus universalis LE-M21 |
| 15 Planktophila versatilis MsE-18 | 44 Allorhodofera lacus MsH-M24 | 73 Methylopusillus universalis MaE-M17 |
| 16 Fimbriococcus planktonicus MsE-15 | 45 Allorhodofera lacus ZE-M1 | 74 Methylopusillus universalis MH-M4 |
| 17 Leadbetterella lacustris RE-19 | 46 Hahnella aquatica MiE-11 | 75 Methylopusillus universalis MH-M5 |
| 18 Flavobacterium rarum TH-M1 | 47 Hahnella lacustris MsE-M47 | 76 Methylopusillus universalis MiE-M1 |
| 19 Flavobacterium neuenschwanderi GE-10 | 48 Lacustriabacter communis MsE-M52 | 77 Methylopusillus universalis RE-M9 |
| 20 Caulobacter lacus MiH-16 | 49 Zwartia planktonica RE-10 | 78 Methylopusillus universalis RH-M37 |
| 21 Aquidulcibacter rimovensis RH-10 | 50 Zwartia lucis GE-14 | 79 Methylopusillus universalis ZE-M6 |
| 22 Aquidulcibacter miladensis MiH-15 | 51 Methylopusillus planktonicus KE-4b | 80 Methylopusillus universalis ZIE-M21 |
| 23 Fontibacterium abundans MiE-29 | 52 Methylopusillus planktonicus LE-M7 | 81 Methylotenera profunda RE-M3 |
| 24 Fontibacterium medardense ME-17 | 53 Methylopusillus planktonicus MaE-M16 | 82 Methylotenera profunda RH-M32 |
| 25 Rhabdaerophilum aquaticum MsE-M23 | 54 Methylopusillus planktonicus MaE-M22 | 83 Methylotenera hypolimnetica RH-M31 |
| 26 Tabrizicola rara LH-M10 | 55 Methylopusillus planktonicus MoH-M15 | 84 Methylotenera hypolimnetica ZIE-M10 |
| 27 Allotabrizicola aquatica RE-M30 | 56 Methylopusillus planktonicus MoH-M17 | 85 Novimethylotenera aquatica ME-M6 |
| 28 Sphingobium cupriresistens MiE-4 | 57 Methylopusillus planktonicus MoH-M19 | 86 Novimethylotenera aquatica MsE-M29 |
| 29 Sphingorhabdus rara RE-M21a | 58 Methylopusillus planktonicus MoH-M36 | 87 Verrucolacustris abundans MiH-22 |

Supplementary Fig. 17: Average nucleotide identities (ANI) between strains from our culture collection and MAGs and previously cultivated species. The species border (95% ANI) is indicated with a solid line. Raw data is given in Supplementary Data 14.

Line 270: Shouldn't it be "Fig. 3B" here?

RESPONSE: This was indeed a typo, thank you for identifying it (now Fig. 4b).

Line 280: Couldn't more TAA and less TGA/TAG simply reflect the general tendency of streamlined genomes to have lower GC-content?

RESPONSE: This is certainly true, we have modified the text accordingly to (L 253-256): “Genome-streamlined strains tended to use a different stop codon (TAA) indicative of nitrogen limitation⁵², which is also reflected in a low GC content, and contained more membrane transporters (normalized per Mbp genome size) than strains with medium to large genomes.

Line 284: “significance distinction” could be reformulated to stress in what way these genomes were distinct (generally smaller/larger).

We are sorry for this confusion and changed the text to clarify that that microbes with genes encoding motility (both flagellar motility and gliding motility) and secretion systems type II, IV, or VI were larger in genome size than those lacking these traits. We changed the text to (L 258-262): “Further, microbes containing genes encoding motility and secretion systems type II, IV, or VI had significantly larger genome sizes (Supplementary Fig. 18, Supplementary Data 4, 18), except for two *Zwartia* strains with large genomes (3.6 and 4.5 Mbp) that lacked genes for flagella assembly and chemotaxis, congruent with previously described *Zwartia* strains⁶⁷.”

Lines 288-312: These two paragraphs are repeated. (I.e., these lines are almost exactly the same as lines 263-287; the only difference between the two versions is the figures and supplementary material).

RESPONSE: Thank you for identifying this mistake, it happened in the last round of revision before submission when we decided to shuffle some parts of the text. The duplicated paragraph has been deleted.

Line 360: Should be “Fig. 4”

RESPONSE: Thank you for finding also this typo, we changed it accordingly (now Fig. 5).

Line 395: Was hypolimnion defined as deeper than 5m?

RESPONSE: The hypolimnion was defined based on thermal stratification (or - in the case of unstratified spring samples - based on previous data for the sampled lakes). We also took care not to sample anoxic water layers.

Line 438: Since viability is a crucial statistic for the paper, it would be preferable to reiterate the formula in this study.

RESPONSE: We included the formula in the methods as suggested (L 391-394): “Isolation success expressed as Viability (V), i.e., probability that a cell selected at random is viable was calculated based on the formula by Button et al.⁴⁹ as follows:

$$V = \frac{\ln(1 - p)}{X}$$

Where p is the number of wells or cultivation tubes, n , with growth z ($p = z/n$) and X is the estimated number of cells inoculated per well.”

Reviewer #4 (Remarks to the Author):

RESPONSE: Thank you very much for your review. We highly appreciate the Nature Communications initiative to facilitate training in peer review for Early Career Researchers.

REVIEWER COMMENTS

Reviewer #1 (Remarks to the Author):

Thanks for your thoughtful responses to my suggestions and congratulations on a very nice study. I will endorse acceptance for publication, but if possible consider changing "72 distinct genus or lineage-like taxa" to something like "72 distinct genera or unnamed SILVA lineages".

- Brian Hedlund

RESPONSE: Thank you very much for your supportive review with helpful comments that largely improve our manuscript. We changed “72 distinct genus or lineage-like taxa” to “72 distinct genera or alphanumerical SILVA lineages” (Line 113).

Reviewer #3 (Remarks to the Author):

The comments raised by us have all been well addressed in the revised version of the manuscript.

RESPONSE: Thank you very much for your review. Your previous comments helped to improve our manuscript.

Reviewer #4 (Remarks to the Author):

RESPONSE: Thank you very much for your review. We highly appreciate the Nature Communications initiative to facilitate training in peer review for Early Career Researchers.